# High-resolution serum proteome trajectories in COVID-19 reveal patient-specific seroconversion

Philipp E Geyer[1,*] ⓘD, Florian M Arend[2], Sophia Doll[1], Marie-Luise Louiset[2], Sebastian Virreira Winter[1] ⓘD, Johannes B Müller-Reif[1] ⓘD, Furkan M Torun[1] ⓘD, Michael Weigand[2], Peter Eichhorn[2], Mathias Bruegel[2], Maximilian T Strauss[1], Lesca M Holdt[2], Matthias Mann[3] ⓘD & Daniel Teupser[2,**] ⓘD

## Abstract

A deeper understanding of COVID-19 on human molecular patho-physiology is urgently needed as a foundation for the discovery of new biomarkers and therapeutic targets. Here we applied mass spectrometry (MS)-based proteomics to measure serum proteomes of COVID-19 patients and symptomatic, but PCR-negative controls, in a time-resolved manner. In 262 controls and 458 longitudinal samples of 31 patients, hospitalized for COVID-19, a remarkable 26% of proteins changed significantly. Bioinformatics analyses revealed co-regulated groups and shared biological functions. Proteins of the innate immune system such as CRP, SAA1, CD14, LBP, and LGALS3BP decreased early in the time course. Regulators of coagulation (APOH, FN1, HRG, KNG1, PLG) and lipid homeostasis (APOA1, APOC1, APOC2, APOC3, PON1) increased over the course of the disease. A global correlation map provides a system-wide functional association between proteins, biological processes, and clinical chemistry parameters. Importantly, five SARS-CoV-2 immunoassays against antibodies revealed excellent correlations with an extensive range of immunoglobulin regions, which were quantified by MS-based proteomics. The high-resolution profile of all immunoglobulin regions showed individual-specific differences and commonalities of potential pathophysiological relevance.

**Keywords** biobanking; biomarker; immunoglobulins; individual-specific; SARS-CoV-2

**Subject Categories** Biomarkers; Microbiology, Virology & Host Pathogen Interaction; Proteomics

See also: **D Memon et al** (August 2021)

## Introduction

The pandemic associated with the severe acute respiratory coronavirus type 2 (SARS-CoV-2) has spread around the globe with massive impact on humankind. By now, coronavirus disease 2019 (COVID-19) has infected and killed millions (https://covid19.who.int/). Thanks to the tremendous efforts of the global scientific community, the virus has been extensively investigated, and new tests for pathogen detection and potential treatments have been rapidly developed (Wiersinga *et al*, 2020).

The clinical presentation of COVID-19 is characterized by a variety of symptoms (Wiersinga *et al*, 2020). The most common manifestations are fever (89%), cough (58%), and dyspnea (45%) (Rodriguez-Morales *et al*, 2020). This is mirrored by rather non-specific laboratory findings, such as decreased albumin, elevated C-reactive protein (CRP), and lymphopenia, which are also commonly seen in other viral diseases. A rather characteristic feature of COVID-19, particularly in severe cases, is venous thromboembolism, which occurred in up to 59% of patients in an intensive care unit setting (Middeldorp *et al*, 2020). On a mechanistic level, dysregulated platelets and neutrophils cooperate to drive a systemic prothrombotic state, indicating inflammation as a trigger for thrombotic complications. As an important laboratory finding, the fibrin degradation product d-dimer was strongly elevated in COVID-19 and correlated significantly with disease severity (Nicolai *et al*, 2020). Another hallmark of COVID-19 is the formation of virus-specific antibodies, which peaked within 3 weeks after symptom onset (Long *et al*, 2020; Buchholtz *et al*, 2021). While currently available routine laboratory tests give important diagnostic cues and have contributed to a better understanding of the pathophysiology, they only provide an incomplete picture of humoral changes in COVID-19.

Proteins control and execute the vast majority of biological processes, and specific alterations in protein levels typically accompany disease onset and progression. Mass spectrometry (MS)-based proteomics is the method of choice to globally investigate proteins in a biological system—its proteome (Aebersold & Mann, 2016). In

---

1   OmicEra Diagnostics GmbH, Planegg, Germany
2   Institute of Laboratory Medicine, University Hospital, LMU Munich, Munich, Germany
3   NNF Center for Protein Research, Faculty of Health Sciences, University of Copenhagen, Copenhagen, Denmark
    *Corresponding author. Tel: +49 176 2332 7838; E-mail: geyer@omicera.com
    **Corresponding author. Tel: +49 89 4400 73210; E-mail: daniel.teupser@med.uni-muenchen.de

---

this sense, MS-based proteome analysis of plasma and serum is unbiased and in principle an ideal technology for systems-wide characterization of disease response (Geyer *et al*, 2017). In practice, body fluid proteomics is very challenging but continuous technological improvements have led to a resurgence of interest (Geyer *et al*, 2017; Ignjatovic *et al*, 2019; Suhre *et al*, 2021).

Several groups have analyzed the serum or plasma proteome of COVID-19-infected patients (D'Alessandro *et al*, 2020; Messner *et al*, 2020; Park *et al*, 2020; Shen *et al*, 2020; Shu *et al*, 2020). These were generally small-scale studies with single or few time points. As a general trend, the levels of complement components and inflammation proteins tended to increase, whereas proteins of the coagulation cascade tended to decrease when compared to control groups. One study investigated a relatively large number of plasma samples in a longitudinal study design to develop predictive models but also reported alterations linked to inflammatory response, metabolic reconstitution, and immunomodulation (Demichev *et al*, 2021).

The aim of our study was to use MS-based proteomics to discover new potential biomarkers and provide a better understanding of the underlying pathophysiology of COVID-19. To this end, we set out to measure protein trajectories in unprecedented detail in a longitudinal cohort of COVID-19 patients. This involved plasma proteome profiling of 720 serum proteomes of patients hospitalized with COVID-19 symptoms and controls. To efficiently and rapidly analyze this large sample set, we developed a very robust workflow based on a recently described "clinical grade" liquid chromatography (LC) system (Bache *et al*, 2018) with a novel trapped ion mobility—time-of-flight mass spectrometer (timsTOF) (Beck *et al*, 2015; Meier *et al*, 2018). This allowed the characterization of 60 serum proteomes per day. The study design followed our recently proposed "rectangular strategy", where samples are measured in as great a depth as routinely possible, and biomarker patterns are extracted from the entire study population (Geyer *et al*, 2017). This further allowed the assessment of sample or analysis quality issues such as contamination with blood cells or coagulation (Geyer *et al*, 2019). Furthermore, aggregated into global correlation maps, the data identify co-regulated factors, physiological processes and enable integration with other clinical results (Albrechtsen *et al*, 2018; Geyer *et al*, 2019; Ignjatovic *et al*, 2019). We previously noted that the levels of most plasma proteins are specific to an individual, making longitudinal studies particularly informative. As each individual serves as its own control, this effectively corrects for inter-individual variations, increasing the likelihood to discover true regulations of protein levels (Geyer *et al*, 2016a; Dodig-Crnković *et al*, 2020).

In this work, we first describe differences between the serum proteomes of COVID-19 patients and those with apparent COVID-19 symptoms who were PCR-negative. We then derive highly detailed time-resolved disease trajectories of serum proteins with on average 15 time points, covering up to 54 days in blood sampling. This study design allowed us to investigate various aspects of the host response to COVID-19 infection as reflected in differences between disease trajectories, longitudinal protein changes, and immunoglobulin production. We disentangle these with global correlation maps that also include detailed clinical chemistry parameters. In particular, the cohort had measurements with five different anti-SARS-CoV-2 immunoassays against antibody classes, enabling us to inspect correlations between these immunoassays and the corresponding MS-detected serum proteins. Our results show that protein levels follow complex patterns and suggest that biomarker tests would benefit from incorporating individual timelines and individual-specific protein levels. We discuss implications of our measurements for our understanding of the antibody-based and individual responses of COVID-19.

## Results

### Study overview and serum proteome analysis

To draw a detailed picture of the dynamic nature of circulating proteins in response to COVID-19, we investigated longitudinal blood serum samples of 31 COVID-19 patients, as well as single time point samples of 262 SARS-CoV-2 PCR-negative controls (Fig 1A).

Patients presented at the University Hospital of the Ludwig-Maximilian University (LMU) Munich with COVID-19-like symptoms. Among a total of 720 samples, 458 were from the 31 COVID-19 patients with an average of 14 samples (7–30) per individual over an average period of 31 days (14–54; Fig 1B and C; Dataset EV1).

Applying recent technological progresses of streamlined MS-based proteomics and an automated sample preparation procedure allowed protein digestion and peptide purification of 720 study samples within a single working day (Fig 1D) (Geyer *et al*, 2016b) (Materials and Methods). For peptide separation, we used an Evosep One LC system in which peptides are first immobilized on a small volume of disposable C18 tip material without carryover, eluted into preformed gradients, and finally separated on a relatively short and robust analytical column with minimal overhead between injections. Mass analysis used the PASEF acquisition principle on a

**Figure 1. Study overview and serum proteome analysis.**

A Overview of the study cohort, including 262 SARS-CoV-2 PCR-negative control patients with single time point samples and 31 COVID-19 patients with longitudinal samples collected during the period of hospitalization.

B Total numbers of samples within each study group.

C Longitudinal trajectories of the covered time in days (*x*-axis) and the number of available samples (*y*-axis) for each patient.

D Automated MS-based proteomics pipeline starting with 1 μl of serum, LC-MS instrumentation to generate MS raw data and data analysis.

E In total, 502 proteins were quantified in this study, covering more than five orders of magnitude of MS signal. Examples of clinically applied biomarkers are labeled.

F Violin plots representing the numbers of quantified proteins in individual serum samples of PCR-negative controls and COVID-19 positive patients. The dashed lines indicate the median, and the dotted lines indicate the quartiles.

G Quality assessment of each sample according to main contamination sources of serum (Geyer *et al*, 2019). Intensities of samples with contamination indicators above a designated cutoff are highlighted in red, and the numbers of samples exceeding these levels are displayed.

H Cross-correlation of quantitative protein levels across all 720 proteomes. Longitudinal samples within individuals are arranged in consecutive order along the axes. A zoom-in of the framed area is depicted in Fig 4E.

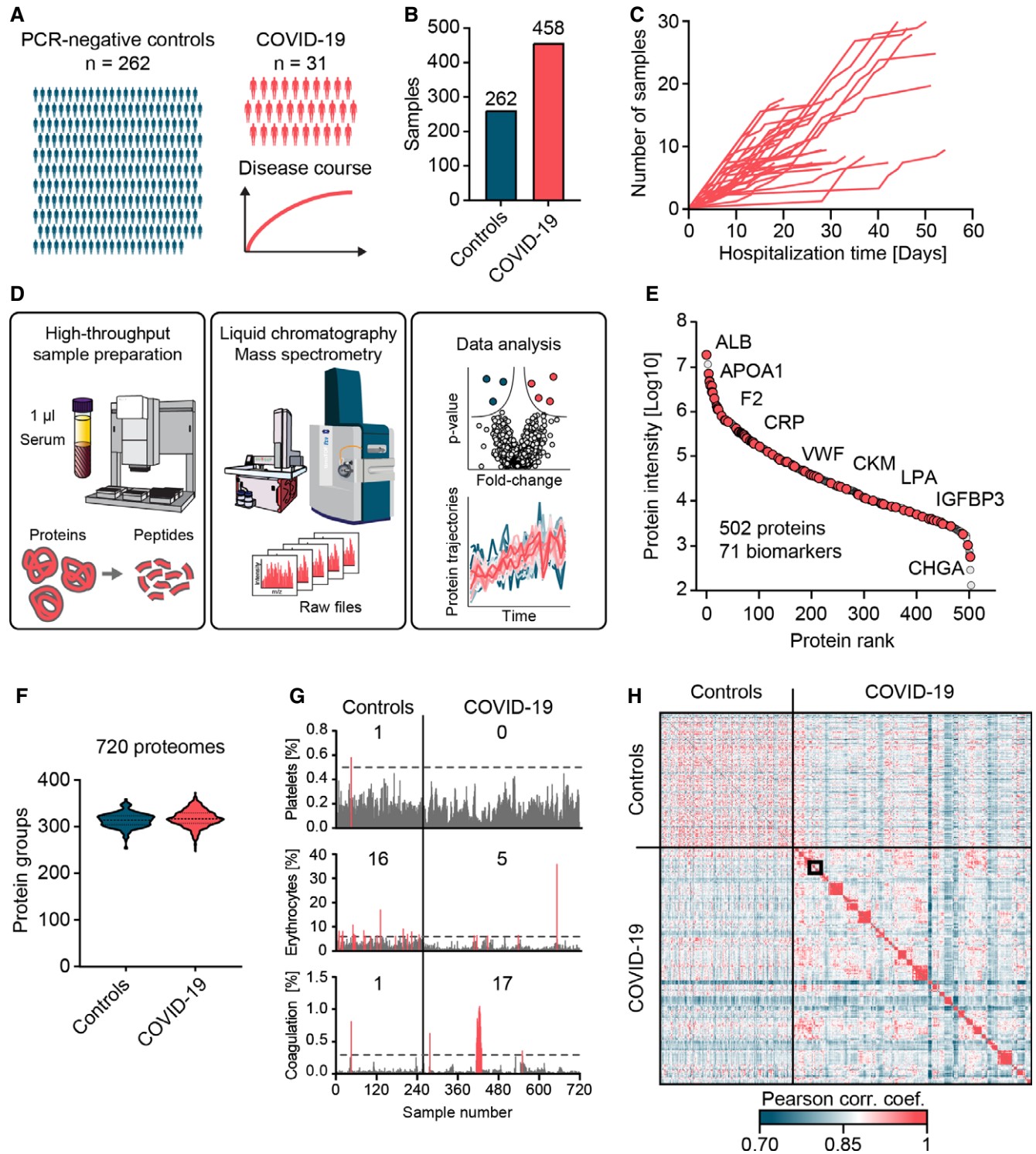

**Figure 1.**

timsTOF instrument, enabling very high sequencing speed and therefore data completeness (Meier *et al*, 2015, 2018; Bache *et al*, 2018). Across all 720 samples, we quantified a total of 502 proteins (Fig 1E). The median number of quantified proteins in samples from COVID-19 positive patients and PCR-negative controls in this rapid method was 312 (± 18) and 308 (± 16), respectively (Fig 1F). The

dataset contained 71 clinically applied biomarkers for a wide range of indications (Dataset EV2). To obtain further insights into the tissues of origin for each protein, we annotated proteins according to the Human Protein Atlas (HPA) based on transcriptomics data of organs (Dataset EV2). In total, 123 proteins were enriched in their expression according to the mRNA data in one specific organ

(Materials and Methods) with the liver as the main origin with 92 proteins. We also annotated proteins for a wide variety of biological functions. Moreover, we highlighted all proteins showing dependencies of age, sex or weigh loss (Dataset EV2).

As a first analysis step, we assessed the quality of all samples according to our previously established quality marker panels in order to pinpoint samples with potential issues in pre-analytical processing. One sample was contaminated with platelets, 21 had evidence of erythrocyte lysis, and 18 had signs of impaired coagulation (Figs 1G and EV1) (Geyer *et al*, 2019). Furthermore, we detected increased erythrocyte protein contaminations in the control group compared to the COVID-19 patient samples (5 vs. 21 samples). Upfront knowledge of these issues turned out to be important as it allowed us to highlight these proteins as potential sources of bias in our further analysis (Fig EV2A–C). As intra-individual variation is expected to be smaller than inter-individual variation, we used a correlation of the 720 proteomes to each other for a global consistency check. Indeed, the large majority of longitudinal samples showed higher correlation within than between individuals (Fig 1H, Appendix Fig S1, see below).

### Serum proteome differences of COVID-19 patients and SARS-CoV-2 PCR-negative controls with COVID-19-like symptoms

Our hypothesis was that alterations of serum protein levels specific to COVID-19 might enable the differentiation of COVID-19 patients from patients with COVID-19-like symptoms. This was the basis for collecting samples of SARS-CoV-2 PCR-positive and PCR-negative patients. The latter presented with COVID-19-like symptoms such as fever, cough, shortness of breath, throat pain, loss of smell and taste, fatigue, general malaise, gastrointestinal complaints, headache, cognitive impairment, need of oxygen, or intensive care treatment because of respiratory symptoms.

Comparing the serum proteomes between the two groups on the first day of sampling resulted in 37 proteins with significantly altered levels of which 14 showed increased and 23 decreased levels in COVID-19 patients (Figs 2A and EV2A, Dataset EV3). Proteins increased in COVID-19 patients included typical innate immune system mediators such as complement factors C2, C9, C4BPA, alpha-1-acid glycoprotein 1 (ORM1), monocyte differentiation antigen CD14, and galectin-3-binding protein (LGALS3BP). Plasma protease C1 inhibitor (SERPING1) was the most significantly regulated protein ($P$-value: $1.7*10^{-11}$; 1.5-fold), and CD14 was the protein with the highest fold-change ($P$: $2.4*10^{-10}$; 2.1-fold) in COVID-19 patients compared to PCR-negative controls. Moreover, a group of protease inhibitors, including SERPING1, SERPINA3, SERPINA10, ITIH3, and ITIH4, were increased in COVID-19 patients. Likewise, coagulation factor V (F5) was significantly increased in COVID-19 patients, whereas modulators of coagulation such as the beta-2-glycoprotein 1 (APOH), histidine-rich glycoprotein (HRG), and fibronectin (FN1) were decreased. Proteins of the lipid homeostasis system, especially components of high-density lipoprotein (HDL) particles such as APOA1, APOA2, APOA4, APOC1, APOD, PLTP, and LCAT were also significantly decreased in COVID-19 patients. APOH was the most significantly regulated of these ($P$: $2.5*10^{-16}$; 1.9-fold), and the cysteine-rich secretory protein 3 (CRISP3) was the protein with the highest fold-change with decreased levels in COVID-19 patients ($P$: $1.1*10^{-4}$; 2.9-fold). Proteins differentially abundant between both groups confirm several findings from former studies, especially the protease inhibitors, proteins of lipid homeostasis, and factors of the immune system (D'Alessandro *et al*, 2020; Messner *et al*, 2020). We also replicated the regulation of gelsolin (GSN), which has been highlighted in previous MS-based proteomics studies. However, GSN ranked at position 34 and was only borderline significant on the first day of sampling ($P$: $1.6*10^{-4}$; 1.3-fold).

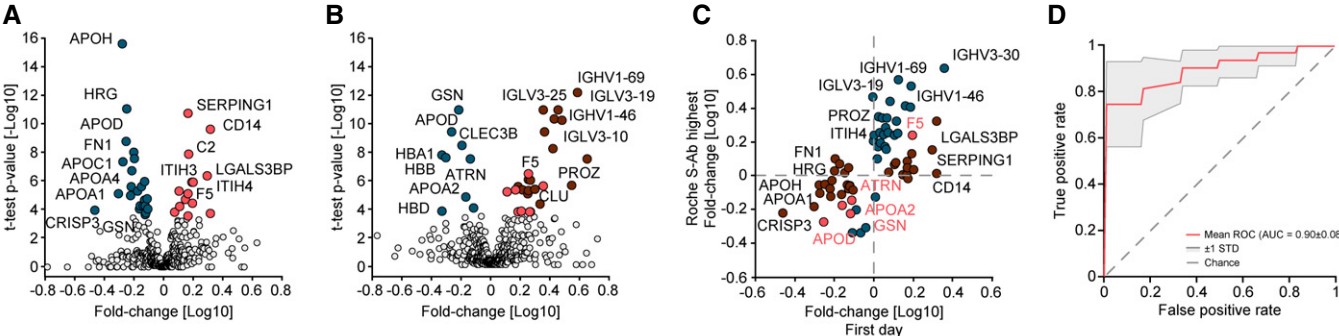

**Figure 2. Serum proteome differences of COVID-19 patients and SARS-CoV-2 PCR-negative controls with COVID-19-like symptoms.**

A  Volcano plot comparing the serum proteomes of 31 COVID-19 patients on the first day of sampling to those of the 262 PCR-negative controls. Significantly up-regulated proteins in COVID-19 positive patients are highlighted in red and down-regulated proteins in blue. Highlighted proteins are significant after multiple hypothesis testing. The $\log_{10}$ fold-change in protein levels is represented on the x-axis and the $-\log_{10}$ t-test P-value on the y-axis. Examples of significantly altered proteins are labeled.

B  Volcano plot comparing the serum proteomes in samples from COVID-19 patients at the time point of highest Roche S-Ab levels to PCR-negative controls. Significantly up-regulated proteins in COVID-19 positive patients are highlighted in red and down-regulated proteins in blue. Significantly up-regulated immunoglobulin regions are highlighted in dark red. Examples of significantly altered proteins are labeled.

C  Scatter plot of protein fold-changes in (A) vs. those in (B). Significant proteins of (A) are highlighted dark red, those of (B) in blue, and significant in both in bright red. Examples of significantly altered proteins are labeled.

D  ROC curve to classify whether a sample was obtained from a COVID-19 positive or a PCR-negative control patient. The mean ROC curve is displayed in red and ± 1 standard deviations are illustrated in gray. The model achieved an area under the curve (AUC) of 0.90.

To investigate alterations of the serum proteome of COVID-19 patients at a later time point to PCR-negative controls, we leveraged the extensive antibody testing that had been performed in our study. We selected the sample of each patient that had the highest level of SARS-CoV-2 antibodies based on the Roche S-Ab test. Comparison at this point, which occurred on average 21 ($\pm$ 12) days after first sampling, resulted in 34 significantly different proteins (Fig 2B and EV2B, Dataset EV4). At this later time points, in contrast to the first one, GSN had the most significantly decreased levels ($P$: $1.2*10^{-11}$; 1.7-fold), which illustrates the importance of the investigated point of time and the present dynamics in COVID-19. In contrast, the group of coagulation system proteins described above was not significantly different at this time point. HDL particle proteins were consistently lower also at this time point in COVID-19 patients, among which APOD ($P$: $4.2*10^{-10}$; 1.9-fold) and APOA2 ($P$: $1.6*10^{-5}$; 1.5-fold) were statistically significant. These examples indicate the advantages of investigating the proteome in a longitudinal fashion as distinct proteins were regulated at distinct time points during disease progression.

With a total of 19 out of the 34 differently abundant proteins, immunoglobulins were the group of proteins that showed the most elevated levels in COVID-19 patients. This reflects the antibody test results but at a much more granular level (see below). Only five proteins were significantly different between the comparisons of both the first day of sampling and the day of highest Roche S-Ab levels with the PCR-negative controls (F5, ATRN, GSN, APOD, APOA2; Fig 2C), providing a clear indication of a massive rearrangement of the serum proteome during the course of the disease.

Additionally, we investigated the effect of gender within the cohort on the levels of plasma proteins. Two proteins pregnancy zone protein (PZP) and sex hormone-binding globulin (SHBG) were highly significantly different between women and men ($-\log_{10}$ P-values of 8.0 and 10.5; (Geyer *et al*, 2016b)), and four additional proteins had smaller effects (APOC4, AFM, ORM1, C9). We highlighted all proteins in the comparisons of COVID-19 patients and PCR-negative controls (Datasets EV3 and EV4).

We applied our recently released open-source machine learning platform OmicLearn to our dataset (preprint: Torun *et al*, 2021). A principal challenge was the low number of samples, limiting the statistical significance of the results. To estimate how good our ML classifier distinguishes COVID-19 patients and PCR-negative controls, we used a Stratified K-Fold cross-validation approach ($k = 5$) to classify patients. With this approach, we split the existing data repeatedly into train and test set and applied the ML pipeline. For the training data of each split, we used a decision tree approach to select the 20 most important proteins which are different between the two classes (Fig EV3A–C). Next, we employed an XGBoost-Classifier (Chen & Guestrin, 2016) on the training subset of proteins and then estimated performance values on the remaining test set of each split. Lastly, we averaged the results of all splits, resulting in a receiver operating characteristic (ROC) curve with an average area under the curve (AUC) of 0.90 $\pm$ 0.08 (Fig 3D) and Precision-Recall (PR) curve with an average AUC of 0.92 $\pm$ 0.06 (Fig EV3A). The positive predictive value of the classifier for the presence of COVID-19 was 0.81, while the negative predictive of the absence of COVID-19 in controls was 0.87 (Fig EV3B).

## Regulated serum proteins in the disease course of COVID-19

To understand the degree and nature of serum proteome remodeling during the disease course in infected patients, we performed three statistical analyses on our dataset (Fig 3A). For all comparisons, we considered the first day of sampling as a baseline. We used a one-sample *t*-test, because proteins vary in an individual-specific manner (Geyer *et al*, 2016a; Dodig-Crnković *et al*, 2020).

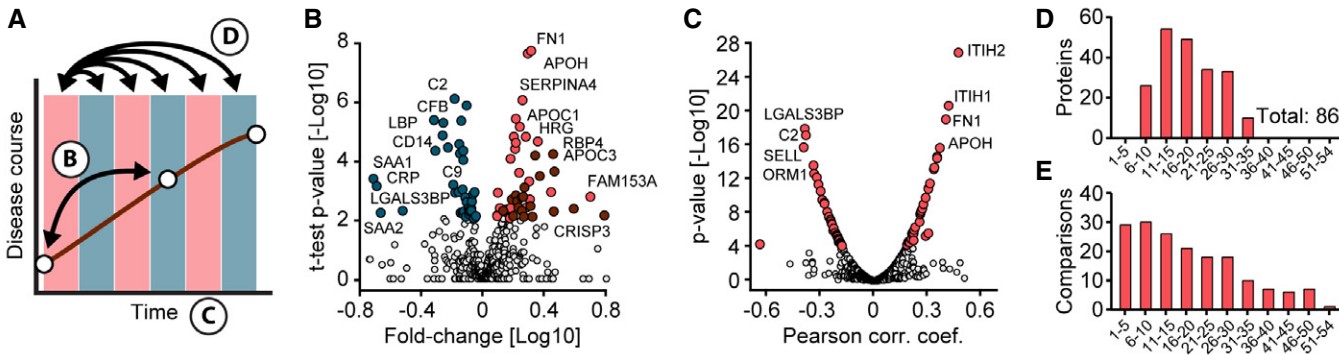

**Figure 3.  Identification of proteins altered in COVID-19.**

A  Schematic of how proteins were compared across disease trajectories. Letters correspond to the panels in this figure. Red and blue boxes indicate binned time intervals.

B  Volcano plot of the results of a one-sample *t*-test comparing the sample on the first day of sampling and the time point with the highest antibody levels. Blue-colored proteins are significantly down- and red ones up-regulated over time. Immunoglobulins are highlighted in dark red. The fold-change in protein levels is depicted on the *x*-axis and the -log10 *t*-test *P*-value on the *y*-axis. Examples of significantly altered proteins are labeled.

C  Correlation of proteins with sampling time during hospitalization. Pearson correlation coefficients and -log10 *P*-values are displayed on the *x*- and *y*-axes, respectively. *Z*-scored protein intensities were used for the correlating to take individual-specific protein levels into account. Proteins significantly correlating with sampling time (positively or negatively) and a *P*-value < $10^{-4}$ are highlighted in red. Examples of significantly altered proteins are labeled.

D  Numbers of significantly altered proteins between the first day of sampling (day 0) and subsequent time intervals (binned days on the *x*-axis), determined by one-sample *t*-tests.

E  Numbers of samples per interval subjected to the one-sample *t*-test in (D).

First, we investigated differences between the first day of sampling (early disease stage) and the time point with the highest host antibody response as determined by the Roche S-Ab assay. This allowed us to anchor the analysis around a clinical parameter specific to each patient. In total, the systemic effects on the serum proteome were accompanied by 38 decreased and 44 increased proteins (Figs 3B and EV2C, Dataset EV5).

The most significantly decreased proteins included the complement factors C2 ($P$: $6.1*10^{-7}$; 1.6-fold) and CFB ($P$: $1.0*10^{-6}$; 1.3-fold), whereas FN1 ($P$: $1.4*10^{-8}$; 2.0-fold) and APOH ($P$: $1.7*10^{-8}$; 1.9-fold) increased most significantly. The median fold-change of significantly regulated proteins was 1.6 from the first day to the day with the highest Roche S-Ab level. As a group, the down-regulated proteins were dominated by factors of the inflammation system, including 18 proteins annotated with the Gene Ontology Biological Process (GOBP) term "immune system process", which included serum amyloid A-1 protein (SAA1; $P$: $3.6*10^{-4}$; 5.2-fold), C-reactive protein (CRP; $P$: $6.3*10^{-4}$; 3.2; 4.9-fold), serum amyloid A-2 protein ($P$: $5.2*10^{-3}$; 4.7-fold), CD14 ($P$: $3.8*10^{-5}$; 2.1-fold), and lipopolysaccharide-binding protein (LBP; $P$: $3.5*10^{-6}$; 2.1-fold). Notably, increased proteins were dominated by immunoglobulins with 20 different regions (see below). In addition to APOH and FN1, the coagulation regulators HRG ($P$: $2.1*10^{-5}$; 1.6-fold), CPB2 ($P$: $1.1*10^{-3}$; 1.2-fold), PROZ ($P$: $3.4*10^{-3}$; 1.9-fold), and TTR ($P$: $4.5*10^{-4}$; 2.0-fold) clearly increased, as did a group of apolipoprotein C proteins (APOC1 ($P$: $5.9*10^{-6}$; 1.7-fold), APOC2 ($P$: $1.3*10^{-5}$; 1.9-fold), APOC3 ($P$: $3.2*10^{-5}$; 1.6-fold)).

Second, we explored regulations of serum protein levels over time, which revealed 34 highly significant positively and 39 highly negatively correlated proteins ($P$-value $< 10^{-4}$; Fig 3C, Dataset EV6). Proteins showing the highest positive correlation were ITIH2 ($P$: $6.4*10^{-27}$; Pearson correlation $R$: 0.47), ITIH1 ($P$: $7.8*10^{-21}$; $R$: 0.42), APOH ($P$: $5.3*10^{-16}$; $R$: 0.37) and FN1 ($P$: $2.8*10^{-19}$; $R$: 0.40). Proteins showing the highest negative correlation were LGALS3BP ($P$: $3.6*10^{-18}$; $R$: −0.39), C2 ($P$: $2.0*10^{-17}$; $R$: −0.38), L-selectin (SELL; $P$: $4.7*10^{-16}$; $R$: −0.40), and ORM1 ($P$: $5.8*10^{-14}$; $R$: −0.34). These data demonstrate a strong, coordinated response of the serum proteome over the time course of infection.

Third, to investigate more complex protein trajectories in COVID-19 infection, we conducted one-sample $t$-tests across 5-day intervals (Fig 3D). This resulted in 28 significantly regulated proteins (Fig 3D, Dataset EV7) in addition to those correlating with sampling time alone. Interestingly, after binning samples in 5-day intervals, there were no significant changes from day 0 to days 1–5 and the first significant changes were detected when comparing to days 6–10.

Strong regulations were found between day 0 and the later intervals up to the days 26–30 (Fig 3D and E). The coagulation-associated proteins described above were again the most significantly increased ones, but the binned time course analysis added AZGP1 and KNG1 to this group. The most significantly decreased proteins consisted of complement factors such as C8A, C8B, CFB, C2, and C9 in the 11–15 and 16–20 days intervals and other inflammation proteins such as ORM1, ORM2, LBP, CD14, LGALS3BP, and CRP in the later intervals.

Together, these three time-resolved analyses (Fig 3B–D) implicate that a large proportion of the quantified serum proteins (130 out of 502) in diverse biological processes are significantly altered in the course of COVID-19. Reassuringly, 52 (79%) of the 66 proteins significantly altered between COVID-19 patients and PCR-negative controls changed over the time course. This highlights both the extensive rearrangement of central physiological proteins and that these can be assessed by proteome profiling.

### High-resolution trajectories and clusters of differentially regulated proteins in COVID-19

Following the analysis of COVID-19 cases against controls as well as binned and grouped time intervals described above, we inspected protein trajectories of all COVID-19 patients in high resolution. Reducing the time span to a maximum of 37 days (for which we had at least five patients per sampling day) resulted in 116 proteins with significant changes along the trajectories. This revealed three major clusters: (i) broadly decreasing, encompassing 51 proteins, (ii) broadly increasing (35 proteins), and (iii) broadly increasing, followed by a decrease (30 proteins; Fig 4A). As the $Z$-scored trajectories substantially overlapped, Fig 4B shows all of them in the form of a heatmap, preserving full resolution of all proteins and time points.

For biological interpretation of regulated proteins, we tested the 116 regulated proteins with keywords and GOBP, Molecular Function (GOMF), and Cellular Component (GOCC) using a Fisher exact test. This resulted in 409 significant associations between the keyword and GO term categories, corresponding to 51 keywords, which were further reduced to 20 non-overlapping terms (Materials and Methods, Dataset EV8). Enzymatic activity was one of the main reported regulations indicated by proteins with keywords "Proteases", "Protease inhibitors", "Zymogen" and "Hydrolases". The keyword "Protease" had the highest number of annotations (25 proteins), reflecting the regulation of plasma protease inhibitors, coagulation factors and the complement system. Other frequent annotations included "Transport" (21 proteins), followed by "Immunity" (19 proteins),

**Figure 4.  Longitudinal trajectories and extent of proteome alterations in COVID-19.**

A  Longitudinal trajectories of the 116 proteins that significantly changed over the disease course of up to 37 days and were quantified in at least five of 31 COVID-19 patients. Trajectories were clustered into three main groups by Euclidian distance after $Z$-scoring and were color-coded by distance from the cluster center to highlight outliers (blue).

B  Longitudinal protein trajectories in COVID-19 over the sampling time of up to 37 days represented as a heatmap and clustered as in (A).

C  Main keywords associated with regulated proteins. Keywords were identified in Fisher exact test on all 116 proteins and then subjected to hierarchical clustering (Materials and Methods).

D  Scatter plot of the serum proteomes of one patient on the first day of sampling (day 0, $x$-axis) compared to day 2 (left panel) and day 18 (right panel). Proteins of the clusters 1, 2, 3 described above are highlighted in blue, dark red, and bright red, respectively. The inflammation markers CRP and SAA1 are labeled.

E  Color-coded Pearson correlation coefficients for all samples of the same patient as in (D). The panel is a zoom-in of the framed area in Fig 1H.

F  Longitudinal variation of the serum proteome for all 31 COVID-19 patients. Pearson correlation coefficients were calculated between the first (day 0) and each consecutive sampling day as shown by example in (D). The trajectory of the median Pearson correlation coefficient is highlighted in red.

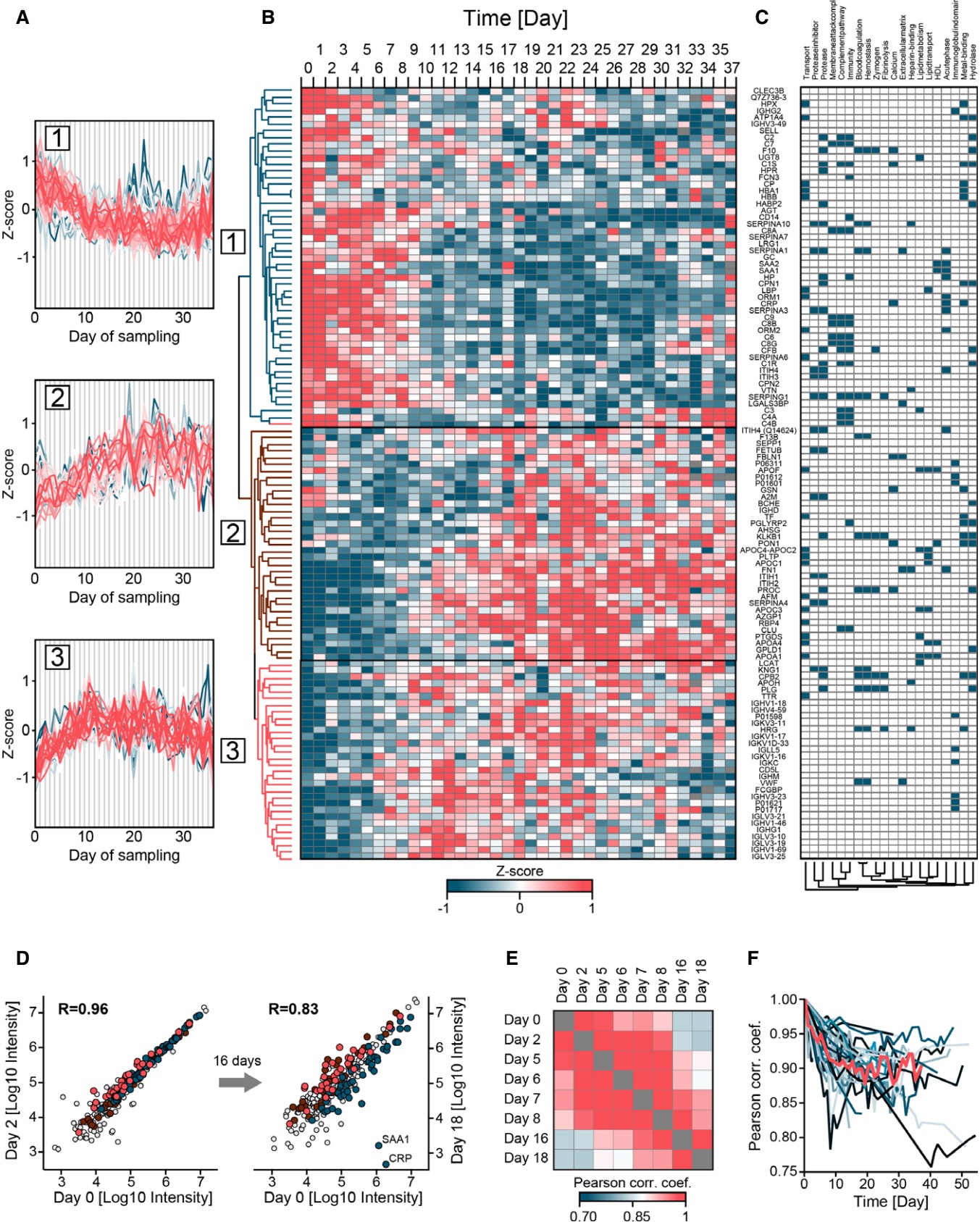

**Figure 4.**

"Complement-pathway" (15 proteins), "Metal-binding" (15 proteins) and "Blood-coagulation" (12 proteins; Fig 4C).

Applied to the three clusters of temporally regulated proteins, the results of Fisher's exact test revealed that cluster 1 contained proteins of the innate immune system such as CRP, SAA1, CD14, ORM1, ORM2, LBP, 13 different complement factors and LGALS3BP (Fig 4B, Dataset EV9). These reflect down-regulation of the immune system in the course of hospitalization at the level of individual proteins. Cluster 2 contained proteins associated with lipid homeostasis, including APOA1, APOA4, APOC1, APOC2, APOC3, PON1, and PLTP. Moreover, proteins involved in coagulation such as FN1, F13B, and K-dependent protein C (PROC) were also increased and members of this cluster. Cluster 3 revealed partly similar trajectories as cluster 2 but decreasing levels toward the later time points. It also contained coagulation-associated proteins such as APOH, VWF, HRG, and several proteases with functions in the regulation of blood coagulation such as kininogen-1 (KNG1), plasminogen (PLG), and carboxypeptidase B2 (CPB2). The joined trajectories of coagulation-associated proteins have been further confirmed by clustering the longitudinal trajectories according to physiological processes (Appendix Fig S2A and B). This is in line with previous reports of coagulopathies (in particular over-activity of this system) as one of the main complications in COVID-19 but adds a temporal and molecular dimension. With 20 proteins, immunoglobulins constitute the largest group in cluster 3.

In order to explore the extent of proteome changes over time on an individual patient basis, we calculated Pearson correlation coefficients of serum proteomes between the first (day 0) and all consecutive sampling days. This is shown by example for one person from day 0 to day 2 and day 0 to day 18 (Fig 4D). As expected, correlation between day 0 and 2 is higher than between day 0 and 18 (0.96 vs. 0.83). The scatter plots also confirm that this divergence is different for the three clusters according to their overall trajectories. In particular, CRP and SAA1 are members of cluster 1 and their levels decreased up to 1,000-fold over time. Integrating these binary proteome comparisons for a typical individual patient reveals a remarkable stability of the individual serum proteomes on consecutive days, while changes over more than a week are much more substantial (Fig 4E). Finally, the Pearson correlation coefficients for all patients over time decrease from a median correlation of 0.96 on day 1 to 0.88 from day 10 on (Fig 4F).

Focusing on the 25 patients that survived COVID-19 infection compared to the six that did not, levels of 14 proteins were different at the last day of sampling. These 14 proteins included factors of the coagulation system such as heparin cofactor 2 (SERPIND1), plasma kallikrein (KLKB1), and PLG. The latter two proteins showed longitudinal alterations in cluster 3, emphasizing the importance of the coagulation system in COVID-19. Interestingly, the isoform Q14626 of ITIH4, a pro-inflammatory acute phase protein was significantly increased in patients that did not survive ($P$: $2.8*10^{-6}$; 2.8-fold) already at the first day of sampling, raising the possibility of prospective classification of disease mortality (Fig EV4). Reassuringly, ITIH4 has been confirmed very recently as a potential predictor for disease mortality in COVID-19 (preprint: Völlmy et al, 2021).

## Global correlation map of 720 proteomes

To better understand the overall associations of all 502 quantified proteins with each other and the 19 clinical parameters of our cohort, we generated a global correlation map (Albrechtsen et al, 2018). This consists of the pairwise correlation of 521 items in all 720 samples (135,460 correlation coefficients) that were subjected to unsupervised hierarchical clustering (Fig 5A). This highlighted 21 positively or negatively correlated groups of proteins and clinical chemistry parameters (Dataset EV10).

The inflammation system formed the largest cluster with 71 co-correlated items. CRP values as quantified by a standard clinical chemistry test showed the highest coefficient of correlation ($R$: 0.95) with MS-quantified CRP, providing a positive control for our work-flow (Fig 5B). Next to positive correlations, we also observed anti-correlating clusters of proteins, which reflect partially on study-specific characteristics. For example, the anti-correlation of the inflammation-dominated cluster and the immunoglobulin cluster can be explained by the longitudinal trajectories of both groups, which are in opposite directions (Fig 4B).

Notably, FN1 and APOH, which were among the most significantly different proteins between the first time point of COVID-19 patients and PCR-negative controls and were also longitudinally regulated, fall all into the same main cluster. Furthermore, they clustered with eight proteins connected to blood coagulation: CPB2, F2, F12, F13B, PLG, KNG1, SERPIND1, and KLKB1, further tying coagulation processes to the time course of COVID-19 at the systems-wide level. The second largest cluster of 52 items was dominated by immunoglobulins and consisted of several strongly co-regulated sub-cluster of antibodies, containing specific immunoglobulin regions. Note that our MS-based proteomics work-flow does not de-novo sequence each antibody, but readily distinguishes antibody classes from each other based on peptide sequences of constant domains.

The cohort was extensively tested by five different SARS-CoV-2 antibody immunoassays, which grouped very closely together. However, they did not fall into the immunoglobulin cluster on the global correlation map, possibly due to the PCR-negative control patients. In agreement with this hypothesis, a second global correlation map limited to COVID-19-positive patients indeed clustered SARS-CoV-2 antibody immunoassays together with the immunoglobulin area (Fig 5C). Of the 49 proteins with a positive correlation to the Roche S-Ab test, 34 belonged to different antibody classes ("immunoglobulin domains", Fig 5D). A similar number of correlations with immunoglobulin regions was identified for the other SARS-CoV-2 antibody immunoassays (Fig EV5A–E, Dataset EV11).

## Antibody secretion during seroconversion in COVID-19

The five SARS-CoV-2 antibody immunoassays resulted in positive responses—indicating seroconversion—in most but not all COVID-19 patients (Fig 6A). The time courses of all patients show orders of magnitude differences in immune response as indicated by the Roche S-Ab immunoassay (Fig 6B, Appendix Fig S3A–F for all tests). To investigate the association of MS-quantified immunoglobulin regions with the five SARS-CoV-2 antibody immunoassays, we selected the five most significantly correlating serum proteins of each assay. This resulted in nine immunoglobulins and four non-immunoglobulin proteins (FCGBP, PROZ, FN1, ITIH4), whose correlation to each of the test is shown in Fig 6C. The antibody chain Ig kappa chain V-III region CLL (P04207) was the protein with the highest significant correlation to the Roche S-Ab test ($P$: $2.3*10^{-53}$; $R$: 0.67), but showed

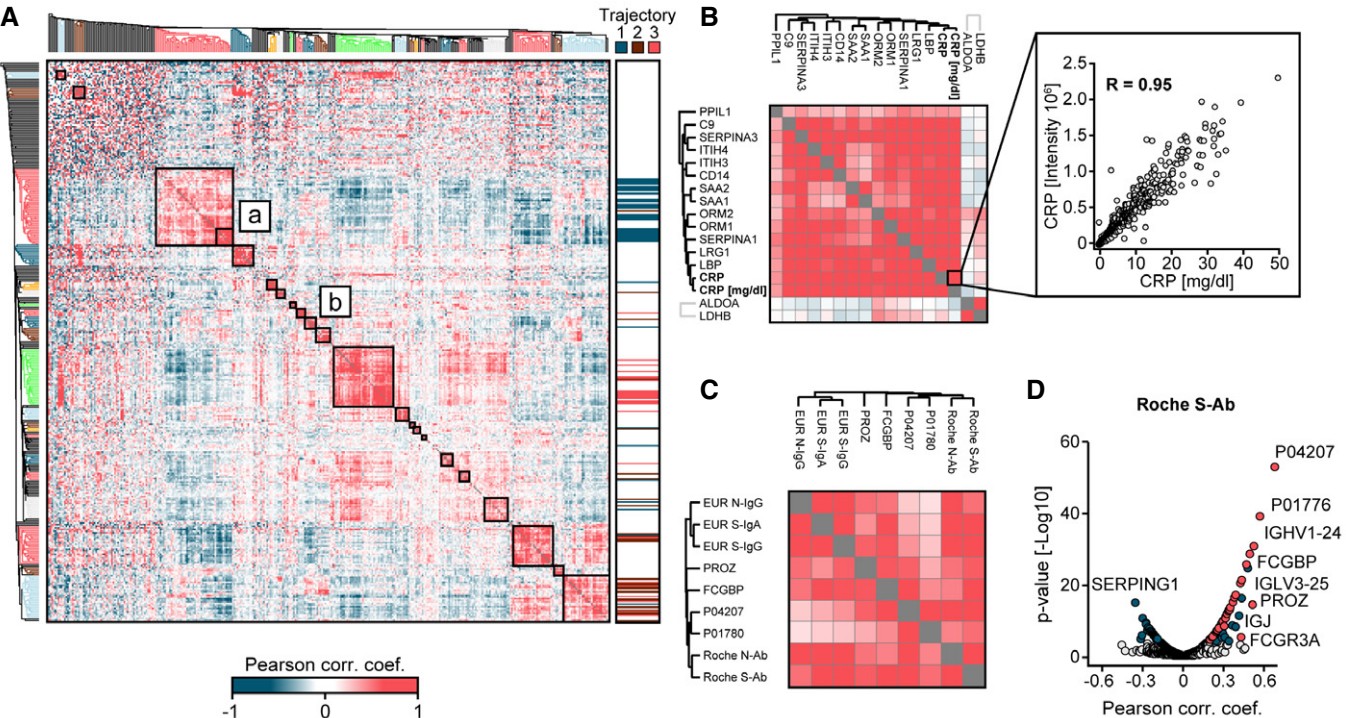

**Figure 5. Clusters of co-regulated proteins from the global correlation map.**

A   Global correlation map of proteins and clinical chemistry parameters based on Pearson correlation coefficients and hierarchical clustering using Euclidean distance. The cluster designated as (a) contains inflammation proteins and (b) proteins correlating with SARS-CoV-2 antibody immunoassay values. Colored sections of the dendrogram highlight clusters of co-regulated proteins. The color code on the right refers to the three trajectories of longitudinally regulated proteins of Fig 4B.

B   Magnification of sub-cluster (a). The zoom-in on the right depicts the correlation between MS-determined CRP and clinical chemistry determined CRP.

C   Magnification of the cluster containing the five SARS-CoV-2 antibody immunoassay values and their correlating proteins from the global correlation map of all COVID-19 patients. The cluster predominantly contains immunoglobulin regions.

D   Proteins correlating with the Roche S-Ab test. Significantly correlated proteins are highlighted in blue and significantly correlated immunoglobulin regions are highlighted in red ($q < 0.05$; $P < 10^{-4}$). Pearson correlation coefficients and *P*-values were calculated, and statistically significantly correlating proteins were determined using a Benjamini-Hochberg FDR correction. Examples of significantly altered proteins are labeled.

little correlation to the EUR S-IgG, EUR N-IgG, and EUR S-IgA tests ($R < 0.30$; Fig 6C). The immunoglobulin J-chain (IGJ; $P$: $2.9*10^{-28}$; $R$: 0.48) and the Ig alpha-1 chain C region (IGHA1; $P$: $1.2*10^{-23}$; $R$: 0.45) highly correlated to the EUR S-IgA assay, which detects immunoglobulin IgA, providing positive control for the serum proteomics measurements. Our unbiased approach clearly associates a large number of specific antibody regions to SARS-CoV-2 infection. Furthermore, it also implicates other proteins, such as FCGBP, which binds constant regions of IgGs and has mainly been described in tissue contexts (Johansson *et al*, 2009).

To investigate how our data resolve individual-specific and protein-specific courses of antibody development, we correlated immunoglobulin regions with SARS-CoV-2 antibody immunoassay levels within each patient. This is exemplified for patient 11, where we separately plot the Roche S-Ab values against the levels of the four most correlating serum proteins. For each of these, seroconversion happened between day 6 and 8 (Fig 6D). While seroconversion always tends to happen within a few days, the time points varied for different patients.

To obtain a global view of the composition of the antibody response detected by the five SARS-CoV-2 antibody immunoassays, we determined the number of significantly correlating

immunoglobulin regions for each test and patient (Fig 6E). With the exception of patients 13, 17, and 22, we found at least one immunoglobulin domain significantly correlating with at least one of the five tests. There were no correlations for patient 17 because none of the antibody tests were significant. Only in seven patients, all five antibody tests were associated with significantly changing immunoglobulins. Interestingly, we found correlating immunoglobulin regions even in cases where the test results had not exceeded the clinical cutoff values (marked by X in the panel). This is illustrated for patient 15, where we found 28 significantly correlating proteins with the EUR S-IgG assay, although this test itself was not above the cutoff (Appendix Fig S4A–D). In total, 24 patients had significant correlations of immunoglobulin regions with the Roche S-Ab test, while only 14 individuals had significant correlations with the EUR S-IgA test (Fig 6F, Appendix Fig S5A–D). The maximum number of correlating immunoglobulins with the Roche S-Ab assay within an individual was 49 (mean: 11). This was not a function of the number of samples per patient (Fig 6F). Significantly correlating immunoglobulins increased on average 4.3-fold, but certain immunoglobulin regions were more than 100-fold elevated.

Next, we investigated the distribution of significantly correlating immunoglobulin regions in the study population. For this purpose,

we counted different immunoglobulins that were significant in each individual. Taking the Roche S-Ab test as an example, the majority of antibody regions were significantly correlated in at least two individuals (Fig 6G). In total, a remarkable 92 out of the 127 quantified immunoglobulin regions were significant, indicating that the large majority of immunoglobulin regions are involved in the response to SARS-CoV-2 infection. Moreover, we identified "favored" antibody regions that were increased in many patients such as IGHV3–15 (12 of 31 patients), IGHV1–69 (11), and IGLV3–10 (11).

Finally, we analyzed the time course of all patients without recourse to the antibody tests. This resulted in a very detailed picture of seroconversion, in which highly correlated immunoglobulin regions clustered closely together (Fig 6H). Even within an individual, the number of immunoglobulin regions, their fold-changes, and time points of increasing levels varied. Note that our MS-based proteomics workflow identifies several peptides per immunoglobulin, sufficient to assign them to immunoglobulin regions while not revealing their complete sequence; hence, the antigen-binding sites are not covered by this analysis. Clusters of trajectories of immunoglobulin regions presumably indicate a common antigen, in this case virus proteins. Interestingly, we observed a general decrease in the levels of specific immunoglobulin regions in several individuals. One of these, the IgM constant domain, reports on the class switch of IgM to IgG directly from the proteomics data.

As expected from the readouts of the SARS-CoV-2 antibody immunoassays (Fig 6B), the immunoglobulin trajectories were highly individual-specific. In patient 11, for example, 33 immunoglobulin regions increased over time, which grouped in two clusters with an average of 4.2- and 2.1-fold increase (Fig 6H). We found that MS-based proteomics provided additional insights in patients with very low SARS-CoV-2 antibody immunoassays values. Although these values were very low in patient 15, we identified two clusters of immunoglobulin regions which increased on average 5.6- and 2.0-fold, allowing this patient's antibody response to be tracked by longitudinal MS-based proteome profiling. Remarkably, in patient 17, who had no positive SARS-CoV-2 antibody immunoassays (but was PCR-positive), two immunoglobulin regions increased two-fold (A0A087WUS7: $P$: $2.6*10^{-4}$; P01708: $P$: $1.9*10^{-4}$) after which their levels stayed elevated. SARS-CoV-2 antibody immunoassays were negative twice and three times just above threshold for

patient 22. MS-based proteomics explained these borderline results as the 53 immunoglobulin regions of this patient dropped on average 5.6-fold compared to the first time points of sampling and indicated a strong effect on the adaptive immune system.

# Discussion

Here, we describe alterations of the serum proteome during COVID-19 in an untargeted manner using a scalable plasma proteome profiling workflow. With a total of 720 serum samples, this is one of the largest MS-based body fluid proteomics efforts, comprising the most detailed longitudinal protein trajectories during hospitalization (average 31 days; maximum 54 days). Furthermore, the comparison of serum proteomes to a control cohort of patients with COVID-19-like symptoms that turned out to be PCR-negative allowed further interpretation of virus-induced alterations.

As a main finding, a quarter of quantified serum proteins (130 of 502; 26%) changed significantly over the disease course, revealing an extensive remodeling of the serum proteome in COVID-19. Confirming this, a study investigating the time course of the plasma proteome during COVID-19 found a comparable portion of protein changes (89 of 309; 29%) (Demichev et al, 2021).

In our study, three clusters of co-regulated proteins with different longitudinal protein trajectories stood out: The first cluster comprised proteins decreasing during hospitalization, the second comprised proteins increasing, and the third cluster comprised proteins increasing within the first 3 weeks and decreasing afterward. Serum proteome changes were striking not only on the cohort level, but just as much at the individual level. The latter could also be seen from the individual trajectories of serum proteome remodeling.

The first cluster of longitudinally altered proteins included predominantly proteins of the innate immune system, which decreased during the first days and remained low, indicating a general decline of the immune system response during hospitalization. This observation has been confirmed by longitudinal studies of inflammatory markers determined by routine clinical chemistry, and immunoassays (Haljasmägi et al, 2020). The global correlation map with an unbiased hierarchical clustering to group proteins with the same regulation across COVID-19 positive and negative patients

---

**Figure 6. Dynamics of SARS-CoV-2 antibody secretion during seroconversion in COVID-19.**

A  Number of individuals with seroconversion as indicated by five different SARS-CoV-2 antibody immunoassays. Dashed horizontal line indicates the total number of patients.

B  Longitudinal trajectories of the Roche S-Ab immunoassay for all COVID-19 patients.

C  For each of the commercial SARS-CoV-2 antibody immunoassays, the top five correlating proteins in the serum proteome were determined, resulting in nine immunoglobulin gene products and four non-antibody proteins (x-axis).

D  Examples of correlations of four different immunoglobulin regions with the Roche S-Ab test for patient 11. Labels on the data points indicate the day of sampling. The data points of early sampling days are clustered near the origin, in contrast to later sample dates where both protein abundance values and Roche S-Ab values are many-fold increased, which is consistent with seroconversion. The Pearson correlation coefficient (R) is displayed.

E  Panels for each COVID-19 patient indicate the number of proteins significantly correlating with each of the five SARS-CoV-2 antibody immunoassays. The X indicates immunoassay measurements not exceeding the cutoff that classifies an individual as having produced antibodies against SARS-CoV-2. Red numbers indicate individuals of which detailed patient-specific trajectories of immunoglobulin regions were highlighted in panel H.

F  Numbers of significantly correlated immunoglobulin regions per patient are shown in a ranked order with red bars (left y-axis). Numbers of samples per patient are shown in gray (right y-axis).

G  Immunoglobulin regions correlating with the Roche S-Ab test in the indicated number of individuals (y-axis). The three immunoglobulin regions significantly correlating with the Roche S-Ab test in the highest number of individuals are labeled.

H  Time-resolved trajectories of consistently quantified immunoglobulins exemplified with patients 11, 15, 17, and 22. Hierarchical clustering grouped trajectories of immunoglobulin regions by Euclidean distance. The columns are arranged by sampling date. The time points are indicated in days above the heatmaps.

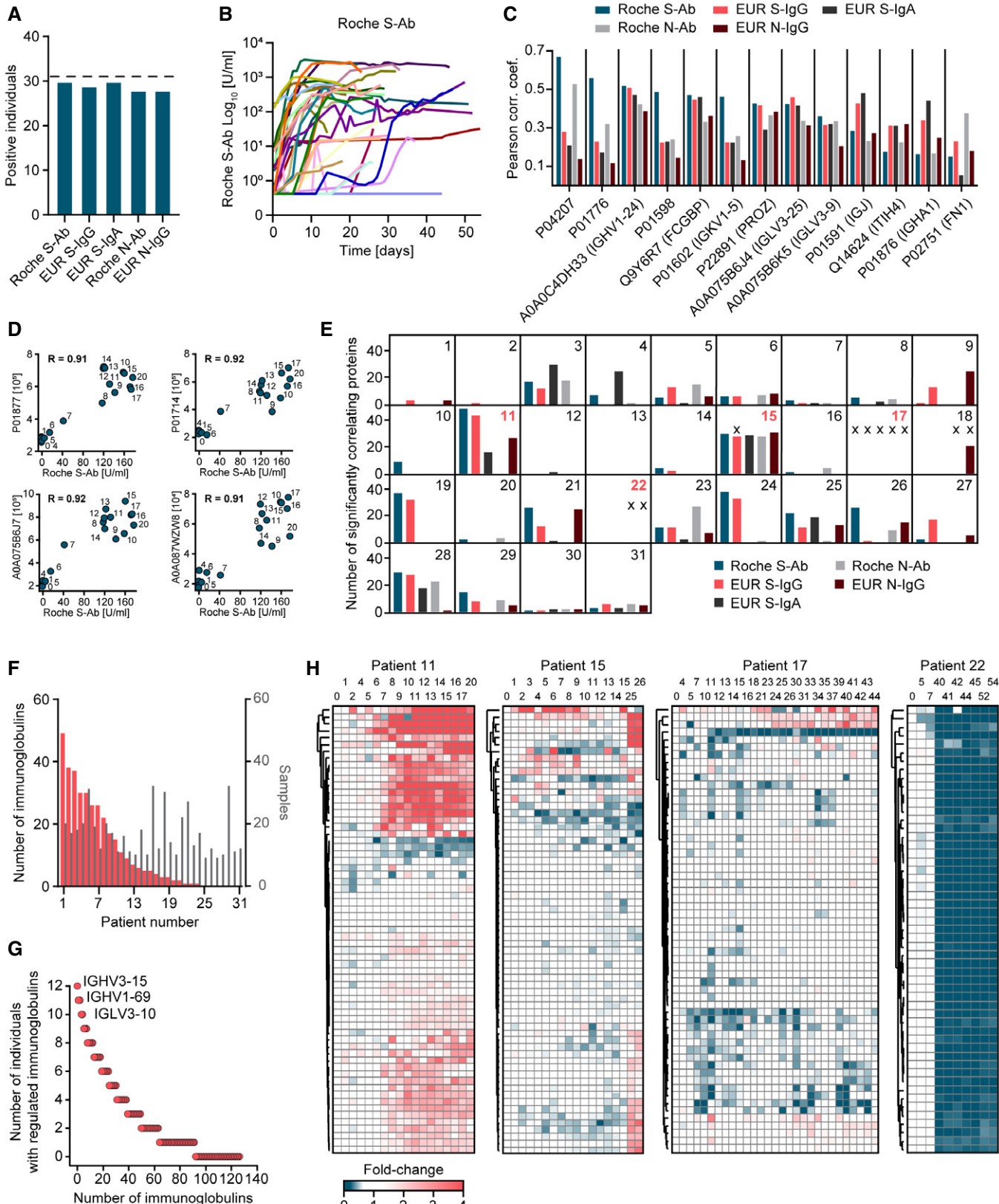

**Figure 6.**

further confirmed the strong systemic effect on the inflammation system. The reaction of the adaptive immune response was indicated by increasing levels of specific antibody regions and SARS-CoV-2 antigen antibodies.

The second cluster, comprising proteins which increased over the course of COVID-19, consisted of proteins associated with lipid homeostasis and coagulation. Changes of proteins related to coagulation and plasma apoprotein levels were corroborated by previous work (Demichev *et al*, 2021). Coagulopathies are a main complication in COVID-19, calling for a detailed understanding of mechanisms of hypercoagulability via identification of proteins regulated in these processes (Demichev *et al*, 2021; Gupta *et al*, 2020; Kollias *et al*, 2020). The description of the detailed regulation of various proteins involved in the coagulation system might even open up the possibility for the development of potential therapeutic avenues.

The third cluster followed a particularly interesting pattern of protein levels, which increased initially and then decreased during hospitalization. This cluster consisted mostly of immunoglobulins, which have not been reported in previous work. Several of the immunoglobulin chains that we found as regulated during seroconversion have more recently been removed from Ensembl and subsequently from the UniProt Knowledgebase, but are still available in UniParc (https://www.uniprot.org/uniparc) (UniProt Consortium, 2018; The UniProt Consortium, 2021). Their inclusion was crucial to our study, which would otherwise have resulted in a much lower number of regulated immunoglobulin regions. More generally, studies of infectious diseases with seroconversion could be applied to confirm and curate public databases. We further extensively characterized the immune response of our cohort by five different immunoassays meant to detect antibodies against the N- and S-antigens of SARS-CoV-2. These were correlated with different types of immunoglobulin regions as quantified by our untargeted proteomics measurements. From the MS-based proteomics data, we further constructed individual-specific time-resolved trajectories of the levels and composition of antibody regions. We found highly individualized responses, but also discovered regions prominently regulated across individuals. Additionally, we found in some patients a disagreement between the quantitative signal of longitudinally regulated immunoglobulin regions identified by MS compared to the SARS-CoV-2 antibody immunoassay measurements. In one case, the SARS-CoV-2 antibody immunoassays resulted in very limited signals; however, MS-based proteomics reported on an increase of a broad spectrum of immunoglobulin regions with fold-changes similar to patients with highly positive responses in the immunoassays. Of note, the different SARS-CoV-2 antibody immunoassays had also a distinct degree of variation in terms of fold-changes and correlations to each other (Buchholtz *et al*, 2021). Hence, the MS readout of the highly detailed immunoglobulin profile could be applied to track seroconversion in patients. In addition to antibody regions, IgGFc-binding protein (FCGBP) prominently correlated with SARS-CoV-2 antibodies. The function of FCGBP is poorly understood and has previously been reported as elevated in serum of patients with autoimmune disease (Kobayashi *et al*, 2001). We found levels of circulating FCGBP to be regulated during seroconversion in our COVID-19 data, and we speculate that it is an indicator of antibody response. This would allow studying the immune response to COVID-19 in a quantitative fashion for each individual and to identify those that produced strong antibody responses and that could serve as donors for production of convalescent serum/plasma therapeutics (Amanat *et al*, 2020).

Adding to longitudinal trajectories, we also performed a comparison of proteomes of PCR-negative controls with COVID-19-like symptoms. This analysis revealed that HRG, FN1, and APOH were among the most significantly regulated proteins showing decreased abundance in COVID-19 patients at the first day of sampling. Our results point toward a complex rearrangement of multiple factors of the coagulation system, in which many of these proteins decrease at earlier time points and increase during disease course toward the levels of PCR-negative controls. Machine learning enabled us to train a classifier that on average correctly identified COVID-19 patients with 81% true positive rate. Conversely, it identified PCR-negative controls with a 87% negative predictive value. These are excellent and promising values given our relatively small cohort.

Our work emphasizes the value of longitudinal study design for biomarker discovery, which allowed to correct for inter-individual variation and determination of proteome alterations in disease progression. Compared to studies with single time points between COVID-19 patients and controls that provided first insights into potentially regulated proteins, our comparison of serum proteomes over the course of disease progression provided a clear set of potential biomarkers which we are now following up in larger cohorts.

# Materials and Methods

### Study cohort

#### COVID-19 patients
Serum samples from 31 COVID-19 patients, admitted to the University Hospital of LMU Munich with acute COVID-19 confirmed by positive PCR, were collected over time from leftover material of samples submitted to the Institute of Laboratory Medicine for routine laboratory diagnostics. Serial samples were collected from each patient, covering a period of up to 54 days from the first day of sampling, adding up to a total of 458 samples. The cohort partially overlapped with a cohort described in our previous work, including the description of the SARS-CoV-2 antibody immunoassays Roche N-Ab (Roche Elecsys Anti-SARS-CoV-2), Roche S-Ab (Roche Elecsys Anti-SARS-CoV-2 S), EUR S-IgG (EUROIMMUN Anti-SARS-CoV-2 (IgG)), EUR N-IgG ( EUROIMMUN Anti-SARS-CoV-2-NCP (IgG)) and EUR S-IgA (EUROIMMUN Anti-SARS-CoV-2 (IgA)) (Buchholtz *et al*, 2021). Clinical and clinical chemistry data were retrieved from electronic patient records. The patients were sampled at both regular wards and intensive care units.

#### PCR-negative control patients
Serum samples from 262 patients, admitted to the University Hospital of LMU Munich with possible symptoms of SARS-CoV-2 but with a negative PCR result, were collected from leftover material of samples submitted to the Institute of Laboratory Medicine for routine laboratory diagnostics. SARS-CoV-2 symptoms included fever, cough, shortness of breath, throat pain, loss of smell and taste, fatigue, general malaise, gastrointestinal complaints, headache, cognitive impairment, need of oxygen, or intensive care treatment because of respiratory symptoms.

Samples were stored as 250 μl aliquots in 2D barcoded biobanking vials (Thermo Scientific, Waltham, Massachusetts, USA) at −80°C in the LMU LabMed Biobank. Anonymized analysis has been approved by the Ethics Committee of LMU Munich (reference number 21-0047). The experiments conformed to the principles set out in the WMA Declaration of Helsinki and the Department of Health and Human Services Belmont Report.

## Sample preparation

Serum samples were prepared for LC-MS/MS analysis as previously published (Geyer *et al*, 2016b). In brief, serum proteins were denatured, alkylated, digested, and peptides purified using an automated liquid handling platform (Agilent Bravo) in a 96 well format. To generate a spectral library, 20 serum samples were pooled and fractioned into 24 fractions using high pH-reversed phase liquid chromatography.

## LC-MS/MS analysis

Digested peptides were separated online via a nanoflow reversed phase chromatography with an Evosep One liquid chromatography (LC) system (Evosep). Peptides were separated on an 8 cm × 150 μm column packed with 1.9 μm ReproSil-Pur C18-AQ particles (Dr. Maisch) using the 60 SPD method with a gradient length of 21 min. The Evosep One was coupled online to a timsTOF Pro mass spectrometer (Bruker Daltonics). The instrument was operated in the DDA PASEF mode with 10 PASEF scans per acquisition cycle and accumulation and ramp times of 100 ms each. Singly charged precursors were excluded, the "target value" was set to 20,000 and dynamic exclusion was activated and set to 0.4 min. The quadrupole isolation width was set to 2 Th for $m/z < 700$ and 3 Th for $m/z > 800$.

## Data analysis

Mass spectrometry raw files were analyzed by MaxQuant software, version 1.6.17.0, and MS spectra were searched against the reference proteome FASTA file, downloaded from https://www.ebi.ac.uk/reference_proteomes/ in January 2020.

A contaminant database generated by the Andromeda search engine and the human database were configured with cysteine carbamidomethylation as a fixed modification and N-terminal acetylation and methionine oxidation as variable modifications. We set the false discovery rate (FDR) to 0.01 for protein and peptide levels with a minimum length of seven amino acids for peptides and the FDR was determined by searching a reversed sequence database. Enzyme specificity was set as C-terminal to arginine and lysine as expected using trypsin and LysC as proteases. A maximum of two missed cleavages were allowed. All proteins and peptides matching the reversed database were filtered out. The mass tolerance used for the main search of each precursor was set to 20 ppm and the minimum number of peptides needed for the quantification of a protein was set to 1.

## Bioinformatics analysis

Bioinformatics analyses were performed in Jupyter notebooks using Python and with the Perseus software of the MaxQuant computational platform. Two-sample *t*-tests were performed for the comparison of different groups. For two-sample tests, we used a two-sided Student's *t*-test and used a permutation-based FDR (0.05) for multiple hypothesis testing. Two-sample tests were performed to identify protein level differences of PCR-negative patients with COVID-19-like symptoms and COVID-19 patients at the first day of sampling and at the time point with the highest Roche S-Ab test response.

## Longitudinal alterations of protein levels

To identify proteins correlating with the time course, we first Z-scored proteins within each individual to take individual-specific protein levels into account. Pearson correlation coefficients were calculated for correlation analysis and a Benjamini-Hochberg FDR correction was applied for multiple hypothesis testing.

One-sample *t*-tests were applied to identify longitudinally altered protein levels between two time points. First, the difference of protein levels between both time points was calculated on $\log_{10}$ transformed data to take individual-specific protein levels into account. This was performed to calculate the difference of the first day of sampling and the sample with the highest Roche S-Ab test response.

One-sample *t*-tests were also applied to identify longitudinally altered protein levels between the first day of sampling and binned time intervals. For this purpose, we normalized the protein levels by referencing to the first day of sampling to take individual-specific protein levels into account. Next, we averaged the normalized values for 5-day intervals (day 1–5, 6–10, ...) and applied a one-sample *t*-test to identify proteins significantly different between the first time point and the median of the intervals. A Benjamini-Hochberg FDR (0.05) has been applied for multiple hypothesis testing.

## Quality assessment

The evaluation of sample quality has been performed according to recently described quality marker panels (Geyer *et al*, 2019). In short, summed intensities of each of three quality marker panels for erythrocyte lysis, platelet contamination, and coagulation have been calculated in addition to the intensities of all non-quality-associated proteins. The percentage of the intensities of the quality marker panels compared to non-quality-associated proteins was calculated to determine the contamination of each sample. If the percentage of erythrocyte protein intensities compared to the total proteome was > 6%, a sample was flagged as having increased erythrocyte proteins. If the percentage of platelet protein intensities compared to the total proteome was > 0.5%, a sample was flagged as having increased platelet contamination. If the percentage of fibrinogen chains was > 0.3%, a sample was flagged as having impaired coagulation. Coagulation of serum samples was impaired according to the quality marker panel in 17 samples of COVID-19 patients compared to just one control sample. A total of 15 out of the 17 samples of COVID-19 patients originated from the same individual.

Potential bias between groups was assessed by highlighting the three quality marker panels in a volcano plot of the comparison of the two groups (Fig EV2). Potential bias was indicated in the text, if present. Outliers in statistical tests that serve as candidates within this manuscript were assessed for potential co-correlations with

platelet and erythrocyte markers. As MS-based proteomics is unbiased in the selection of proteins for evaluation, we report on a broad scope of information that we can use to evaluate potential outlier proteins in more detail. Herein, we used the quality marker panels for further evaluation of statistically significant outlier. The quality marker panel indicated a bias toward increased erythrocyte proteins in controls, which was reflected in the results comparing COVID-19 positive and negative patients with significant proteins of typical erythrocyte proteins such as the hemoglobin chains HBA1, HBB, and HBD and the bias of quality marker proteins in one side of the *t*-test (Fig EV2A–C). In the same vein, we reported previously (Geyer *et al*, 2019) that GSN can be an indicator for platelet contamination. However, other platelet markers were not enriched in controls and a global correlation analysis revealed that GSN was not co-regulated with other platelet markers in this study, confirming that GSN changed due to COVID-19 infection. Of all significantly regulated proteins, only hemoglobin chains clustered within quality marker panels, indicating that they originated from erythrocytes.

## Protein trajectories

Significantly longitudinally regulated proteins were defined as proteins that have a statistically significant difference between the first time sampling time points and other time points. Samples were available for each COVID-19 patient at the first time point (TP 0), but not at every other time point. To increase the statistical power for the identification of longitudinally changing proteins, we binned proteomes over a distinct time window of always 5 days: 1–5, 6–10, 11–15, 16–20, 21–25, 26–30, 31–35, 36–40, 41–45, 46–50, 51–54. For this purpose, we calculated the difference within each individual from the first sample to all other time points and calculated the median according to the above listed time windows. Next, we applied a one-sample *t*-test, which resulted in 86 statistically significant proteins for all comparisons (Dataset EV7). Next, we selected all proteins that were statistically significant in one of the above-mentioned tests to identify longitudinally changing proteins, resulting in 130 proteins. The protein intensities were *Z*-scored within each individual over time. We calculated the median of the *Z*-scores for each time point for which we had at least five samples, resulting in 37 time points and 116 proteins fulfilling this criterion. The median *Z*-scores of the proteins were then subjected to a hierarchical clustering with Euclidean distance.

## Intra-individual proteome remodeling

The proteome remodeling was assessed by calculating Pearson correlation coefficients between the proteome at the first time point and the other time points. The median Pearson correlation coefficient plot was calculated only for time points with samples of at least five individuals, hence, covering up to 37 days.

## Keyword annotation of regulated proteins

Keywords and GOBP, GOCC, and GOMF terms were added to the 116 proteins. A Fisher´s exact test was applied between the Keywords and the GO terms. This resulted in 409 significant associations from 51 keywords. Keywords "3D structure", "Completeproteome", "Referenceproteome", "Polymorphism", "Directproteinsequencing", "Repeat",

"Secreted", "Signal" and "Disulfidebound" were excluded due to the general nature of the terms. The keywords "Secreted" and "Signal" were combined to "Secreted", "Innateimmunity" and "Immunity" to "Immunity", "Serineprotease" and "Protease" to "Protease", "Serinproteaseinhibitor" and "Proteaseinhibitor" to "Proteaseinhibitor", "ImmunoglobulinVregion" and "Immunoglobulindomains" to "Immunoglobulindomains", "Complementalternatepathway" and "Complementpathway" to "Complementpathway", "Cytolysis" and "Membraneattackcomplex" to "Membraneattackcomplex" due to their high similarity. The 20 Keywords which had the most significant associations with a GO term were selected for Fig 4A. The complete list can be found in Dataset EV8.

## Tissue-specific proteins

Proteins were annotated for organ-specific expression according to the Human Protein Atlas (HPA) (https://www.proteinatlas.org/). Organ-specific categories of proteins in the HPA are based on transcriptomics data, defining "enriched" proteins with at least four times higher mRNA levels in one organ compared to any other tissue. Group-enriched proteins have at least four-fold higher average mRNA levels in a group of 2–5 tissues compared to any other tissue. Tissue enhanced proteins have at least four-fold higher mRNA levels in a particular tissue compared to the average level in all other tissues. The origin of each protein determined in this way is supplied in Datasets EV2–EV7.

## Weight loss related proteins

Proteins changing due to weight loss were classified according to our previous weight loss study (Geyer *et al*, 2016a) and highlighted in Datasets EV2–EV5.

## Correlation analysis

We calculated Pearson correlation coefficients of binary comparisons of proteins and/or clinical parameters. We applied a hierarchical clustering on top of the correlation matrix using Euclidean distance. Based on the clustering, 21 groups of co-regulated proteins and/or clinical parameters were identified. To draft correlation plots (U-plots), the correlation of clinical to proteomics data was done with Python version 3.8.5. using Pandas (1.1.3), Numpy (1.19.2), Scipy (1.5.2), and Statsmodels (0.12.0) packages. In brief, Pearson correlation coefficients and Pearson *P*-values of protein intensity values to other numerical parameters were calculated. To address multiple testing, Benjamini–Hochberg FDR was employed for *P*-value correction.

## Machine learning

We used OmicLearn (1.0.0) for performing the data analysis, model execution, and generating the plots and charts (preprint: Torun *et al*, 2021). Within OmicLearn, machine learning was done in Python (3.8.8). Feature tables were imported via the Pandas package (1.0.1) and manipulated using the Numpy package (1.18.1). The machine learning pipeline was employed using the scikit-learn package (0.22.1). For generating the plots and charts, Plotly (4.9.0) library was used. Data were normalized in each split using a

**The paper explained**

**Problem**
The COVID-19 pandemic has spread around the globe with massive impact on humankind. A deeper understanding of the molecular pathophysiology of the disease is urgently needed as a foundation for the discovery of new biomarkers and therapeutic targets.

**Results**
A total of 458 proteomes from longitudinal serum samples of 31 hospitalized COVID-19 patients and proteomes from serum samples of 262 controls with COVID-19-like symptoms but not infected with SARS-CoV-2 were analyzed by mass spectrometry (MS)-based proteomics, covering 502 serum proteins. This revealed an extensive number of 116 proteins (26% of all quantified proteins) significantly changing over the course of the disease. Especially, proteins of the innate immune system, coagulation, and lipid homeostasis were altered. A machine learning model was able to identify a set of 20 proteins, differentiating between COVID-19 patients and controls. Moreover, a potential prospective marker of COVID-19 mortality (ITIH4) was identified. In addition, individual-specific trajectories of immunoglobulin regions could be applied to monitor seroconversion in patients.

**Impact**
Serum proteomes over the course of COVID-19 provide a better understanding of the disease pathophysiology, including a time-resolved picture of altered serum protein levels and the functional association between proteins, biological processes, and clinical chemistry parameters. A set of 20 biomarkers could aid in identifying COVID-19 infections, and ITIH4 might serve as potential marker for disease mortality.

StandardScaler approach. To impute missing values, a median imputation strategy is used. Features were selected using an ExtraTrees ($n\_trees = 100$) strategy with the maximum number of 20 features. Normalization and feature selection were individually performed using the training data of each split. For classification, we used a XGBoost-Classifier (random_state = 23 learning_rate = 0.3 min_split_loss = 0 max_depth = 6 min_child_weight = 1).

**Individual-specific immunoglobulin trajectories**

Immunoglobulin regions were filtered for 100% valid values within each individual. Next, the time points were sorted from the first to the last day of sampling. A hierarchical clustering based on Euclidean distance was applied to group similar trajectories together. In patient 17, a two-sample *t*-test was performed to compare protein intensities of P01708 and A0A087WUS7 between the earlier (0–24 days) and later time points (25–44 days).

# Data availability

The MS raw data and MaxQuant output files of the searches generated during and/or analyzed during the current study are available from the corresponding authors on reasonable request.

**Expanded View** for this article is available online.

## Acknowledgements
We thank in particular Britta Pauli and Babett Rannefeld for their expert technical assistance and Peter V. Treit for editing and discussions. We further thank Elisabeth Gasteiger from the UniProt Consortium for the assessment and discussion of immunoglobulin sequences of the UniProt Knowledgebase. We also thank the entire team at OmicEra Diagnostics GmbH for helpful input and discussion. The work carried out in this project was funded by OmicEra Diagnostics GmbH and the University Hospital of LMU Munich, and supported by the German Federal Ministry of Education and Research (BMBF) project ProDiag (grant numbers: 01KI20377A, 01KI20377B) and the German Biobank Alliance, Munich (01EY1711C).

## Author contributions
PEG, FMA, LMH, MM, and DT designed the experiments, supervised and guided the project, discussed proteomics and clinical data, and wrote the manuscript. PEG, SD, MTS, SVW, FMT, JBM-R, MM, and DT analyzed and interpreted MS-based proteomics data. PEG and JBMR performed and interpreted the MS-based proteomics analysis, did bioinformatics analysis generated text and figures for the manuscript. FA, M-LL, MW, PE, and MB interpreted proteomics and clinical chemistry data, discussed these findings, and generated text.

## Conflict of interest
Competing financial interests: PEG, SVW, SD, MTS, FMT, and JBMR are employees of OmicEra Diagnostics GmbH. The other authors declare no conflict of interest.

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
