## [Review Process File · EMBO Molecular Medicine]

High-resolution serum proteome trajectories in COVID-19 reveal patients-specific seroconversion

Philipp Geyer, Florian Arend, Sophia Doll, Marie-Luise Louiset, Sebastian Virreira Winter, Johannes Müller-Reif, Furkan Torun, Michael Weigand, Peter Eichhorn, Mathias Bruegel, Maximilian Strauss, Lesca Holdt, Matthias Mann, and Daniel Teupser

DOI: [10.15252/emmm.202114167](https://doi.org/10.15252/emmm.202114167)

Corresponding author(s): Philipp Geyer (geyer@omicera.com) , Daniel Teupser (daniel.teupser@med.uni-muenchen.de)

Review Timeline:

Submission Date:	22nd Feb 21
Editorial Decision:	23rd Mar 21
Revision Received:	25th Apr 21
Editorial Decision:	11th May 21
Revision Received:	16th May 21
Accepted:	25th May 21

Editor: Jingyi Hou

Transaction Report:

Thank you again for submitting your work to EMBO Molecular Medicine. We have now heard back from the three referees who evaluated your manuscript. As you will see from the reports below, the referees acknowledge the potential interest of the study. However, they also raise a series of concerns about your work, which should be convincingly addressed in a major revision of the present manuscript.

The referees' recommendations are rather clear, and there is no need to reiterate their comments. Importantly, all three referees pointed out the samples' clinical information is missing, and they requested additional analyses (such as prediction analysis) to further enhance the medical relevance of the presented data.

We would welcome the submission of a revised version within three months for further consideration. Please note that EMBO Molecular Medicine strongly supports a single round of revision. As acceptance or rejection of the manuscript will depend on another round of review, your responses should be as complete as possible.

***** Reviewer's comments *****

Referee #1 (Comments on Novelty/Model System for Author):

In "High-resolution longitudinal serum proteome trajectories in COVID-19 reveal patients-specific seroconversion" Geyer et al. performed MS-proteomics analyses of clinical blood samples obtained from patients with confirmed or suspected COVID19. They referenced the protein levels to longitudinal changes and levels of anti-SARS-CoV-2 antibodies. The manuscript is well written, the data and its analyses were extensive, and the story fairly easy to follow.

Even though I am very positive to seeing this proteomics work and the different types of analyses, I do miss a biomedical conclusion that goes beyond a technical and analytical feasibility statement. Besides presenting maps, associations and trajectories, it remains unclear to me what the main (novel) outcome of the cross-sectional and longitudinal studies would be (as compared to Messner, Park and Shu). A clearer statement about the interpretation of the key biomedical findings (rather than proteomics observations) would increase the impact of the work.

Many of the shortlisted proteins originate from the liver, thus confirm the central role and importance of this organ for the circulating proteome and COVID19 phenotypes, but also indicate some of the limitations of the chosen technology. Thus, lower abundant inflammatory markers remained out of reach. While this is not per se an issue, it limits the provided insights into COVID19. Hence, I recommend to conduct a focussed analysis of the liver/intestinal-centric protein regulation in relation to disease and time, rather than harvesting differential hits from the series of analyses.

To condense some of the issues related to the heterogeneity within the disease phenotypes and how these project into their longitudinal variability, it would be relevant and informative to see how networks of proteins and their relations change over time. WGCNA or other tools could deliver some valuable insights into how the different physiological processes connect with another during the course of the disease. An example could be to focus on the inverted clustering in Fig 4C, where certain process occur more frequently during the earlier, middle and later phase of the disease (centring this around the peak in antibody levels). This would provide a more comprehensive global view on the health states in a heterogeneous population rather than depicting individual proteins that may or may not change in all subjects to the same extend, at the same time and into the direction.

Limited to no clinical data (age, sex, BMI, WBC) and information about medication or treatment were given. There is though clear evidence about the impact and importance of, for example, BMI on COVID19: [https://www.cdc.gov/mmwr/volumes/70/wr/mm7010e4 .htm](https://www.cdc.gov/mmwr/volumes/70/wr/mm7010e4.htm), and proteins discussed here have been recently reported with a causal associations to BMI: <https://www.nature.com/articles/s41467-021-21542-4>. It remains therefore unclear how these clinical traits and parameters influence the baseline data and trajectories. Importantly, it miss a deeper referencing of the observed plasma profiles to other (internal) studies, so it becomes easier

to to understand which proteins appear to be more COVID19 related and which are altered or varying due to other causes (eg loss in BMI).

Longitudinal analyses are challenging as it often remains unclear if linear or non-linear changes are to be expected. Some changes occur due to dietary fluctuations, hour of sampling, or during particular treatments (eg ventilation). A key to our understanding of heterogeneity is to know the (clinically healthy) baseline levels. Protein levels may revert to baseline or stay elevated after the peak of the infection has been reached. If available, severity of symptoms (in addition to antibody levels) should be used as to define the course of the disease. One option in addition to the correlation schemes, is to use linear mixed (effect) models and reference the protein levels to the clinical parameters that change over time. By anchoring the analyses onto these recorded clinical traits and parameters, new informative relationships may/will occur. This would also make the study's findings easier to translate and replicate by others.

In case of the clinical non-COVID19 samples presented here, their use as reference samples without any understanding of the clinical data (eg age, sex, BMI, CRP, WBC, ...) is limiting the value of studying these, as it remains unclear if COVID19 diagnosis or other reasons drive the observed differences.

With the current longitudinal design, a commonly expected question would aim to answer the progression/outcome of the disease based on the first sample taken. Were any proteins enriched for survival, severity, or death? Did any of these proteins differ from the non-COVID19 group?

When seeing the immunoglobulin data, I missed to obtain levels of IgM for the early phases of the infection (prior to the S-Ab peak). Could these be provided or is the Roche S-Ab assay capturing these? Please elaborate on this.

Other comments:

- Did the authors also search their data for the presence of SARS-CoV-2 proteins?
- Avoid the use of "differentially expressed" when discussing levels in a systemic body fluid - use "abundant" or "secreted" instead.
- Please reference all proteins in relation to their (primary) tissue of origin.
- Fig 5A: Please reference all correlations with random pairs of correlation to indicate the margin that related factors actually connect with. Why did the authors only focus on positive and not the highly negative correlation values? Aren't the negative ones more informative as they provide perpendicular (= added) rather than concordant (= confirmatory) insights? It would be very interesting to see negative co-relation.
- Fig 6D: the correlation values are highly biased by the two populations of the data.
- Fig 6G: What is the evidence that the different regions belong to anti-SARS-CoV-2 specific antibodies? Can this be stated as is even without performing pull-down experiments?

Referee #1 (Remarks for Author):

This is a very comprehensive study of a COVID19 related cohort and I am very positive to seeing this published in EMBO Molecular Medicine after performing additional data analyses that will lift this work beyond the currently existing "reports". There is an opportunity to address heterogeneity in disease course and to include variability as a give factor rather making this an observation that

will be seen as a limitation of the outcome.

Referee #2 (Remarks for Author):

In this study Geyer and colleagues perform an extensive proteomic characterisation of the changes in ~300 serum proteins levels for 31 COVID-19 patients over ~30 days which could be compared with serum protein levels of 262 PCR-negative controls. This allowed the authors to study the changes in protein levels between patients and controls as well as the changes in protein levels along the course of the disease. Notable changes include serum proteins linked with innate immunity, regulation of coagulation and lipid homeostasis. By analysing a large number of samples the authors could show that there are clear patterns of co-regulation of the serum proteins across samples, which is expected given that there are multiple groups of functionally linked proteins profiled. Finally they could show that the proteomics results are also capable of reporting on seroconversion with several immunoglobulins and other proteins showing strong correlations with antibody test results and a large number of immunoglobulin regions showing varied patterns of changes in expression.

The study of the immune response and the identification of biomarkers of COVID-19 response continue to be of critical importance. This study is most related with the work by Demichev and colleagues, cited in this manuscript, that has performed a similar analysis of 139 patients, measuring ~300 serum proteins also along the time course of infection. The two studies reach similar conclusions in regards to several of the serum proteins changing upon infection. However, the Demichev study goes further in making use of the data to define biomarkers of response severity. Overall, I find the current study presented here to be very descriptive and not making the best possible use of the data generated (see below). While a prior similar study could sometimes be grounds for suggesting a manuscript for rejection, in this case the work by Demichev et al is itself still under review and more importantly than that, the current study presents an independent cohort that is profiled at very high standards. With some improvements on the analysis I think this work can be of very high value for the study of COVID-19.

Major concerns:

1 - The major concern I have with this study is the very descriptive nature. It would be very important to show that the proteomic data can be used as a predictor of infection outcome. This could include but not be limited to: predicting that a patient is indeed infected (with COVID-19); predicting the severity of the outcome; predicting the time for recovery, etc. For this the authors should make use of the data to build predictors, leaving out some of the their patient data for testing (i.e. machine learning with cross-validation, training/testing). It is important to take into account the characteristics of the patients such as age and other factors that may contribute independently for disease progression.

2 - The patient cohort is not well describe in terms of age, gender, disease progression and clinical markers. As described in point 1, some of this information needs to be considered when building the predictors as these may confound some of the associations found. That is, there may be serum protein markers that relate to age and not necessarily with disease progression. Such information should be also provided in supplementary materials or if there any privacy concerns through some data sharing mechanism that protects the patient information.

3 - In my view, the most important benefit of this dataset and analysis will be as independent

analysis/cohort from that described in Demichev et al. In an ideal world the predictive models and biomarkers developed in one cohort would be then tested with the independent measurements from the other cohort. This would be fantastic also as it would allow the authors to see the impact of measurements done in different labs etc. Unfortunately, the authors of Demichev et al. did not make the data available in the preprint. In an ideal scenario it would be a fantastic service for the community if the authors of this manuscript could reach out to the authors of Demichev et al. to obtain the data for this cohort in attempt to perform a comparative analysis. In the absence of this, it would at least be important to improve the comparison between this manuscript and the one by Demichev et al.

Minor comments

1 - There is not information in supplementary tables regarding the proteomics dataset itself. This needs to include at least the protein intensities collected for each individual.

2 - At many points the article is very descriptive but does not provide much context on why some information is reported. It would be useful to better connect the observed findings with underlying biology context.

Referee #3 (Remarks for Author):

The manuscript by Geyer et al. describes a large scale MS based biomarker discovery study of COVID-19 patient serum samples. The study made use of symptomatic controls as well as longitudinal samples from 31 COVID19 patients over an average of 31 days, and by comparing the proteomes of the two cohorts was able to demonstrate specific alterations to the proteome of COVID-19 patient serum samples. The authors also compared the COVID-19 patient samples longitudinally, revealing three major clusters of co-regulated proteins indicating that proteome undergoes complex reorganization over the course of patient hospitalization. In addition to this, the authors demonstrated that a cluster of proteins mostly composed of different immunoglobulin regions correlated with seroconversion as measured by clinical chemistry assays for 5 anti-SARS-COV-2 antibodies. The longitudinal expression of this cluster appeared to be patient specific, and some of the proteomic changes associated with seroconversion were apparent in patients with otherwise negative clinical chemistry results for SARS-COV-2 antibodies suggesting MS based approaches might be more sensitive. From a technical perspective the study was well conducted utilizing a relatively quick (21 minute method) LC-MS method to analyze 60 samples a day, with an average of approximately 310 proteins quantified by LFQ per sample (502 total), demonstrating the high degree of efficiency and sensitivity afforded by the latest LC-MS instrumentation (Evosep One and Bruker timsTOF Pro operated in DDA PASEF mode). The paper was generally well written, although some sections could use revision to improve accessibility.

- The work is impressive in its scope and is interesting from a mechanistic perspective, however the relevance of the study to the broader medical community is unclear since few associations were made with disease severity, hospital length of stay, outcome etc. With the exception of the brief discussion of ITIH4, no association was made between the proteomics data and outcome in a predictive sense. Additionally there is no association of the proteomic data with any information concerning disease management, or other patient treatment which might affect the interpretation of the results. For example in figure 6H, it would be interesting to know if medical history or treatment could explain the immunoglobulin profile for patient 22.

- Similarly, a large part of manuscript details the complex longitudinal remodeling of the proteome during hospitalization. It is suggested that these alterations correlate with disease progression, but

it would also be interesting to note if any proteins or protein clusters correlate with disease management.

- The qualifications for the statistical significance of a putative biomarker are not clearly defined.
- Additionally there is no attempt at validation of any putative marker using complementary methods (MS or otherwise). In some cases clinical chemistry assays correlated (for example CRP), but it would be interesting to validate some of the interesting putative biomarkers (ITIH4) using a more quantitative approach (for example MRM or PRM with stable isotope labeled internal standard peptides).

Specific points for the author's are detailed below:

- The graphical abstract could be simplified. Why are protein trajectories differentiated from proteome alterations? There is no extensive discussion of patient resolution provided in the manuscript.
- The abstract states that biomarkers are needed for COVID-19, but this should be clarified. What types of biomarkers for example, and which - if any putative markers were found in this study.
- In figure 1C, it looks like most proteins detected are clinically utilized biomarkers based on the color coding, but this is not the case.
- On page 6, for consistency the expression of CRISP3 should be written as COVID-19+ relative to the control group (ie most down-regulated in COVID-19 samples).
- What (if any) fold change cut-off was used for statistical significance, and how was this determined. Additionally are p-values in the volcano plots (for example figure 2) adjusted for FDR?
- For the global correlation map (Fig 5) it is not clear from Table EV9 which are the 19 clinical chemistry parameters, it would be good to have this as a separate list. Reference to figure 5C should also include reference to figure EV3? How was the number of correlation coefficients calculated on page 12 (135,460)?
- In figure 6F, is the x-axis showing the number of patients, or the patient number? If the later could you comment on the apparent trend (decreasing IGs from patient 1 -> 31)
- Regarding the sample preparation on page 18, samples are referred to as plasma, but they were actually serum in this case.
- Regarding the conditions for LC-MS/MS, could you comment on the relatively large precursor isolation widths used? What were the accepted mass error tolerances (precursor and product) for peptide assignment and was there a minimum number of unique peptides needed for protein quantitation?
- Regarding the Data Analysis section on page 18, why are the fixed/variable modifications listed specifically for the contaminant database? Should this be for both the reference and contaminant databases?

Point-by-point answers to Reviewers' comments

We thank the Reviewers for the positive and in-depth evaluation of our manuscript. We greatly appreciate the constructive comments, which helped us to improve our manuscript. In summary, the Reviewers highlighted the cutting edge technology, the large-scale proteomics effort and that the manuscript was well written. In agreement with the Reviewers' comments, we do see one of the largest advantages in a community effort to gain maximum insight into COVID-19 as a disease and how it effects human molecular pathophysiology.

One of the main Reviewers' comments was to supply a detailed overview of patient characteristics and highlight potential dependencies such as on age, sex and BMI. We appreciate this important suggestion and have now included patient characteristics in an additional supplementary Table EV1. Additionally, we investigated the effect of patient characteristics on the serum proteome and highlighted proteins that have a potential relation with age, sex or weight loss in the previously reported supplementary tables reporting on statistical tests. Furthermore, we indicated the tissue origin of all serum proteins in the supplemental tables.

To further address the suggestion of including predictions, we made use of our recently published open-source machine learning platform OmicLearn (www.OmicLearn.com; (Torun *et al.*, 2021)). We used this platform to predict whether a patient has COVID-19 or not from their plasma proteomes. This resulted in a new panel in the main Fig 2 (Fig 2D) and an additional supplemental figure (Fig EV4A-C). To further add medically relevant information, we also supply a new volcano plot for the comparison of individuals who survived COVID-19 and individuals who did not (Fig EV6). Moreover, we added a supplemental figure with a biological process centric analysis of longitudinal trajectories (Fig EV5A/B). We hope that the Reviewers and Editors will find the new data and supplied material insightful and also feel that they strengthen the revised version of the manuscript.

Referee #1 (Comments on Novelty/Model System for Author):

In "High-resolution longitudinal serum proteome trajectories in COVID-19 reveal patients-specific seroconversion" Geyer et al. performed MS-proteomics analyses of clinical blood samples obtained from patients with confirmed or suspected COVID19. They referenced the protein levels to longitudinal changes and levels of anti-SARS-CoV-2 antibodies. The manuscript is well written, the data and its analyses were extensive, and the story fairly easy to follow.

Even though I am very positive to seeing this proteomics work and the different types of analyses, I do miss a biomedical conclusion that goes beyond a technical and analytical feasibility statement. Besides presenting maps, associations and trajectories, it remains unclear to me what the main (novel) outcome of the cross-sectional and longitudinal studies would be (as compared to Messner, Park and Shu). A clearer statement about the interpretation of the key biomedical findings (rather than proteomics observations) would increase the impact of the work.

In our manuscript, we aimed to provide a fundamental description of the effects of COVID-19 on human molecular pathophysiology by investigating longitudinal trajectories of altered proteins in COVID-19 patients during hospitalization, thus laying a foundation for biomarker discovery. A first step for this major task is to know how proteins are differently abundant between cases and controls - the next step is to elucidate the regulation of these proteins over time. In our manuscript, we describe longitudinal trajectories of 116 significantly regulated proteins, find coregulated clusters of proteins and identify the underlying pathophysiological processes that correlate with these trajectories. The main results were a decrease of inflammatory proteins such as CRP, SAA1, CD14, LBP and LGALS3BP early in the time course, and an increase of proteins of the coagulation system such as APOH, FN1, HRG, KNG1, PLG. These results were also highlighted in the abstract.

We agree with the Reviewer that clearer statements should be included. Therefore, we rephrased the introduction of the abstract:

"A deeper understanding of COVID-19 on human molecular pathophysiology is urgently needed as a foundation for the discovery of new biomarker and therapeutic targets."

We also added a statement in the Discussion that the proteins of the coagulation system might be potential therapeutic targets:

"The description of the detailed regulation of various proteins involved in the coagulation system might even open up the possibility for the development of potential therapeutic avenues."

Apart from laying a foundation for biomarker discovery in COVID-19 by supplying insight into pathophysiology, we agree with the Reviewer that a statement towards the relevance of proteins in disease diagnostics would be of interest for the broad medically and scientifically interested readership of EMBO Molecular Medicine:

- We included a prediction of whether a patient has COVID-19 or not by machine learning. For this purpose, we applied our recently released open-source machine-learning pipeline OmicLearn. This resulted in a new panel in the main

Fig 2 (2D) and three panels in the new supplemental Fig EV4A-C (please see below).

Fig 2 - Serum proteome differences of COVID-19 patients and SARS-CoV-2 PCR-negative controls with COVID-19-like symptoms

A. Volcano plot comparing the serum proteomes of 31 COVID-19 patients at the first day of sampling to those of the 262 PCR-negative controls. Significantly up-regulated proteins in COVID-19 positive patients are highlighted in red and down-regulated proteins in blue. Highlighted proteins are significant after multiple hypothesis testing. The log10 fold-change in protein levels is represented on the x-axis and the -log10 t-test p-value on the y-axis.

B. Volcano plot comparing the serum proteomes in samples from COVID-19 patients at the time point of highest Roche S-Ab levels to PCR-negative controls. Significantly up-regulated proteins in COVID-19 positive patients are highlighted in red and down-regulated proteins in blue. Significantly up-regulated immunoglobulin regions are highlighted in dark red.

C. Scatter plot of protein fold-changes in (A) vs. those in (B). Significant proteins of (A) are highlighted dark red, those of (B) in blue and significant in both in bright red.

D. ROC curve to classify whether a sample was obtained from a COVID-19 positive or a PCR-negative control patient. The mean ROC curve is displayed in red and ±1 standard deviations are illustrated in grey. The model achieved an area under the curve (AUC) of 0.9.

Fig EV4 - Prediction whether a samples is from a COVID-19 positive patient or a PCR-negative control

A. The median Precision Recall (PR) curve (red) and PR curves ±1 standard deviations are displayed (grey).

B. Confusion matrix depicting the predicted and actual disease group of patients.

C. Feature importance (weights) from the ML classifier averaged over all cross-validation runs.

- These new analysis resulted in the following paragraph, which we included in the revised Results section:

“We applied our recently released open-source machine learning platform OmicLearn to our dataset (Torun *et al.*, 2021). A principal challenge was the low

number of samples, limiting the statistical significance of the results. To estimate how good our ML classifier distinguishes COVID-19 patients and PCR-negative controls, we used a Stratified K-Fold cross-validation approach (k=5) to classify patients. With this approach, we split the existing data repeatedly into train and test set and applied the ML pipeline.

For the training data of each split, we used a decision tree approach to select the 20 most important proteins which are different between the two classes. Next, we employed an XGBoost-Classifier (Chen and Guestrin, 2016) on the training subset of proteins and then estimating performance values on the remaining test set of each split. Lastly, we average the results of all splits, resulting in a Receiver Operating Characteristic (ROC) curve with an average AUC (Area Under the Curve) of 0.90 ± 0.08 and Precision-Recall (PR) Curve with an average AUC of 0.88 ± 0.10 . The positive predictive value of the classifier for the presence of COVID-19 was 0.87, while the negative predictive value of the absence of COVID-19 in controls was 0.81.”

New Discussion section:

“Machine learning enabled us to train a classifier that on average correctly identified COVID-19 patients with 87% true positive rate. Conversely, it identified PCR-negative controls with a 81% negative predictive value. These are excellent and promising values given our relatively small cohort.”

New Methods section:

“We used OmicLearn (1.0.0) for performing the data analysis, model execution, and generating the plots and charts. Within OmicLearn machine learning was done in Python (3.8.8). Feature tables were imported via the Pandas package (1.0.1) and manipulated using the Numpy package (1.18.1). The machine learning pipeline was employed using the scikit-learn package (0.22.1). For generating the plots and charts, Plotly (4.9.0) library was used. Data was normalized in each split using a StandardScaler approach. To impute missing values, a Median-imputation strategy is used. Features were selected using an ExtraTrees (n_trees=100) strategy with the maximum number of 20 features. Normalization and feature selection was individually performed using the training data of each split. For classification, we used a XGBoost-Classifier (random_state = 23 learning_rate = 0.3 min_split_loss = 0 max_depth = 6 min_child_weight = 1).“

Many of the shortlisted proteins originate from the liver, thus confirm the central role and importance of this organ for the circulating proteome and COVID19 phenotypes, but also indicate some of the limitations of the chosen technology. Thus, lower abundant inflammatory markers remained out of reach. While this is not per se an issue, it limits the provided insights into COVID19. Hence, I recommend to conduct a focussed analysis of the liver/intestinal-centric protein regulation in relation to disease and time, rather than harvesting differential hits from the series of analyses.

We agree that a targeted investigation for proteins with a distinct origin such as liver/intestinal centric will be more focused, however, in our opinion one of the largest

advantages of MS-based proteomics is the unbiased nature of this technology with regards to the observed proteins. Hence, focusing on a subsection of the dataset might not necessarily make the best of this technology. Nevertheless, we agree with the Reviewer that it would be useful for the community to know the origin of the proteins, which we find significantly regulated. Therefore, we supply for all proteins and additional for all statistically significant proteins the organ or tissue origin in the supplemental Tables EV1-EV6 and a statement in the Results and the Materials and Methods sections:

Results:

“To obtain further insights into the tissues of origin for each protein, we annotated proteins according to the Human Protein Atlas (HPA) based on transcriptomics data of organs (Table EV2). In total, 123 proteins were enriched in their expression according to the mRNA data in one specific tissue (Material and Methods) with the liver as the main origin with 92 proteins. We also annotated proteins for a wide variety of biological functions. Moreover, we highlighted all proteins showing a dependencies of age, sex or weigh loss (Table EV2).”

Materials and Methods:

“Tissue specific proteins – Proteins were annotated for organ-specific expression according to the Human Protein Atlas (<https://www.proteinatlas.org/>). Organ-specific categories of proteins in the HPA are based on transcriptomics data, defining ‘enriched’ proteins with at least four-times higher mRNA levels in one organ compared to any other tissue. Group-enriched proteins have at least four-fold higher average mRNA levels in a group of 2-5 tissues compared to any other tissue. Tissue enhanced proteins have at least four-fold higher mRNA levels in a particular tissue compared to the average level in all other tissues. The origin of each protein determined in this way is supplied in Tables EV2-7.”

To condense some of the issues related to the heterogeneity within the disease phenotypes and how these project into their longitudinal variability, it would be relevant and informative to see how networks of proteins and their relations change over time. WGCNA or other tools could deliver some valuable insights into how the different physiological processes connect with another during the course of the disease. An example could be to focus on the inverted clustering in Fig 4C, where certain process occur more frequently during the earlier, middle and later phase of the disease (centring this around the peak in antibody levels). This would provide a more comprehensive global view on the health states in a heterogeneous population rather than depicting individual proteins that may or may not change in all subjects to the same extend, at the same time and into the direction.

We thank the Reviewer for this comment and implemented the suggested analysis of longitudinal trajectories. First, we clustered the keywords based on Fig 4C and sorted the longitudinal protein trajectories accordingly. This worked out well for distinct keywords such as the group of complement factors or the coagulation system. Therefore, we would include the analysis as the new supplemental Fig EV5A/B. However, the resulting map generated a relatively fragmented view of the longitudinal protein trajectories. One of the main reason for this fragmentation seems to be missing annotations. For example, a quarter of all proteins were not annotated with the keywords (most, but not all were immunoglobulins).

Fig EV5 - Physiological process centric longitudinal protein trajectories.

A. Main keywords associated with regulated proteins are highlighted and hierarchical clustering sorts proteins into similar annotated groups.

B. Longitudinal protein trajectories in COVID-19 over a sampling time of up to 37 days represented as a heatmap and clustered as in (A) according to the keyword clustering.

Challenged by the missing information about protein annotation, we set out to do additional analyses. To obtain a clearer idea of the regulation of the proteins of the three longitudinal clusters of protein trajectories in Fig 4B, we highlight them now color-coded in a panel in Fig 5A to indicate the clusters of proteins in which they fall.

Limited to no clinical data (age, sex, BMI, WBC) and information about medication or treatment were given. There is though clear evidence about the impact and importance of, for example, BMI on COVID19: <https://www.cdc.gov/mmwr/volumes/70/wr/mm7010e4.htm>, and proteins discussed here have been recently reported with a causal associations to BMI: <https://www.nature.com/articles/s41467-021-21542-4>. It remains therefore unclear how these clinical traits and parameters influence the baseline data and trajectories. Importantly, it miss a deeper referencing of the observed plasma profiles to other (internal) studies, so it becomes easier to to understand which proteins appear to be more COVID19 related and which are altered or varying due to other causes (eg loss in BMI).

We agree with this Reviewer and the other Reviewers that additional data describing patient characteristics should be supplied. We now added a new Table EV1 of patient characteristics, which shows that COVID-19 patients and PCR-negative controls were balanced for the BMI.

Unfortunately, BMI was not tracked during the disease course. To investigate the potential effect of weight loss on the investigations, we reanalyzed data from one of our previous plasma proteomics studies, in which we investigated a longitudinal sample set from a caloric restriction induced weight loss cohort (Philipp E Geyer *et al.*, 2016). In this study 52 individuals lost on average 12% of their body weight which is more than double the published average weight loss in hospitalized COVID-19 patients (4.4-6.1%) (Bedock *et al.*, 2020; Di Filippo *et al.*, 2020). Next, we compared the fold-changes of significantly altered plasma proteins in the weight loss study with changes observed in COVID-19. We used the time point with the highest antibody levels as a reference for this comparison. If a protein trajectory was strongly confounded by weight loss, the observed effect in the caloric restriction study should be larger than the effect of longitudinal protein changes as the changes in the caloric restriction induced weight loss study were much larger. This was not the case for a single protein.

Nevertheless, 10 out of the 86 altered proteins had significant changes in the same direction as in the caloric restriction study, which means that these proteins could have been affected by potential weight loss to some degree (Figure below illustrates the ratio of protein level changes in the COVID-19 study and the weight loss study). Of note, the effect will be much lower because the weight loss during COVID-19 is less extensive than in the caloric restriction study and because the effect sizes were smaller in the caloric restriction study.

Another 17 proteins with significant changes in the caloric restriction study and in the COVID-19 study were regulated in an opposite direction.

We include the following statement in the manuscript and we report proteins that showed significant alterations in the aforementioned caloric restriction induced weight loss study in supplemental Table EV5.

“Additionally, we investigated the effect of gender within the cohort on the levels of plasma proteins. Two proteins pregnancy zone protein (PZP) and sex hormone-binding globulin (SHBG) were highly significantly different between women and men ($-\log_{10}$ p-values of 8.0 and 10.5; (Philipp E. Geyer *et al.*, 2016)) and four additional proteins had smaller effects (APOC4, AFM, ORM1, C9). We highlighted all proteins in the comparisons of COVID-19 patients and PCR-negative controls (Table EV3/4)”.

Longitudinal analyses are challenging as it often remains unclear if linear or non-linear changes are to be expected. Some changes occur due to dietary fluctuations, hour of sampling, or during particular treatments (eg ventilation). A key to our understanding of heterogeneity is to know the (clinically healthy) baseline levels. Protein levels may revert to baseline or stay elevated after the peak of the infection has been reached. If available, severity of symptoms (in addition to antibody levels) should be used as to define the course of the disease. One option in addition to the correlation schemes, is to use linear mixed (effect) models and reference the protein levels to the clinical parameters that change over time. By anchoring the analyses onto these recorded clinical traits and parameters, new informative relationships may/will occur. This would also make the study's findings easier to translate and replicate by others.

We agree that dependencies and effects will not always be linear, especially over time and that the baseline state of an individual will affect protein levels. We thank the reviewer for

pointing this out, but we also want to highlight that we included these issues already in the previous version of the manuscript, which we might not have made sufficiently clear. We took individual-specific protein levels in this analysis into account by Z-scoring protein intensities within each individual and using the Z-scored protein intensities for the correlation. We stated this now more clearly in the “Materials and Methods” section and in the legend of Fig. 3C:

“Z-scored protein intensities were used for the correlating to take individual-specific protein levels into account.”

We agree with the Reviewer that several precautions and even clinical parameters such as anchor points should be included, which we actually did in the manuscript. The first precaution is already the longitudinal study design. In total, up to 70% of all protein levels might be specific to an individual (Philipp E Geyer *et al.*, 2016; Dodig-Crnković *et al.*, 2020). So this should be taken into account whenever possible in a study. Using a longitudinal study design inherently allows us to correct for the baseline differences in protein levels. We have accounted for this by referencing to the first day of sampling and by using one-sample t-tests. Moreover, to detect proteins with different longitudinal trajectories, we applied different types of analyses.

As suggested by the Reviewer, we anchored one of the analysis around a “biological” reference point. For this purpose, we used the clinical chemistry measurement of the highest antibody levels, quantified by the Roche S-Ab SARS-CoV-2 antibody immunoassay. This resulted in Fig. 3B. We also made this clearer in the Results section:

“First, we investigated differences between the first day of sampling (early disease stage) and the time point with the highest host antibody response as determined by the Roche S-Ab assay. This allowed us to anchor the analysis around a clinical parameter specific to each patient. In total, the systemic effects on the serum proteome was accompanied by 38 decreased and 44 increased proteins (Fig 3B, Fig EV2C, Table EV5).”

Additionally, we binned proteomes in distinct time intervals to identify complex non-linear protein alterations. These were identified by one-sample t-tests and by taking the first time point of sampling as a reference point to adjust for individual-specific protein levels.

In case of the clinical non-COVID19 samples presented here, their use as reference samples without any understanding of the clinical data (eg age, sex, BMI, CRP, WBC, ...) is limiting the value of studying these, as it remains unclear if COVID19 diagnosis or other reasons drive the observed differences.

We now report a more detailed overview of patient characteristics and clinical data in Table EV1.

With the current longitudinal design, a commonly expected question would aim to answer the progression/outcome of the disease based on the first sample taken. Were any proteins enriched for survival, severity, or death? Did any of these proteins differ from the non-COVID19 group?

To add data for the disease outcome analysis, we further supply now a volcano plot for the analysis of the 25 patients that survived COVID-19 infection compared to the six patients that did not (Fig EV6). Interestingly, a recently published longitudinal study investigating the proteomes between patients with fatal outcome and survivors in COVID-19 also highlighted the protein ITIH4 as significantly differently regulated (Völlmy *et al.*, 2021).

Fig EV6 - Volcano plot showing the results of the comparison of 25 patients that survived COVID-19 infection and six patients that did not. ITIH4 was the only protein with a statistically significant difference.

Furthermore, we applied the above-mentioned machine learning algorithms and analysis to predict fatal outcome in COVID-19. These findings are also reported for the same protein. However, the numbers for predictive models especially for the fatal outcome are very small. We are therefore reluctant to include this analysis in the manuscript.

When seeing the immunoglobulin data, I missed to obtain levels of IgM for the early phases of the infection (prior to the S-Ab peak). Could these be provided or is the Roche S-Ab assay capturing these? Please elaborate on this.

The Roche S-Ab and N-Ab assays have a preference for IgG, but both can detect IgM. MS-based proteomics has no preference for the proteins. IgM heavy chains are highlighted by arrows in patients 11 and 15, which are also part of a manuscript figure. For illustration purposes, we also show here patient 25, which has a textbook regulation of IgM with increasing and decreasing levels over three weeks.

Other comments:

- Did the authors also search their data for the presence of SARS-CoV-2 proteins?

We have done this separately, but did not detect any peptides belonging to SARS-CoV-2 proteins.

- Avoid the use of "differentially expressed" when discussing levels in a systemic body fluid - use "abundant" or "secreted" instead.

We changed this in the revised version.

- Please reference all proteins in relation to their (primary) tissue of origin.

Please see above, we have done this according to the Human Protein Atlas.

- Fig 5A: Please reference all correlations with random pairs of correlation to indicate the margin that related factors actually connect with. Why did the authors only focus on positive and not the highly negative correlation values? Aren't the negative ones more informative as they provide perpendicular (= added) rather than concordant (= confirmatory) insights? It would be very interesting to see negative co-relation.

We actually have mentioned the negative correlations already briefly in the initial version of the manuscript, but we have focused on positive correlations as we are unfortunately limited

in the information that we can supply in the story. Nevertheless, we agree with the Reviewer that anti-correlated factors are indeed very interesting and can add further biological insights. We do already supply the whole correlation matrix in the supplemental Table EV10, which will allow the interested reader to investigate correlations and anti-correlations in more detail. To highlight this possibility, we added an example for such an anti-correlation in the main text of the manuscript:

“Next to positive correlations, we also observed anti-correlating clusters of proteins, which reflect partially on study-specific characteristics. For example the anti-correlation of the inflammation dominated cluster and the immunoglobulin cluster can be explained by the longitudinal trajectories of both groups, which are in opposite directions (Fig. 4B).”

- Fig 6D: the correlation values are highly biased by the two populations of the data.

The fold changes between the available time points are quite strong, resulting in the observed correlations. We supply another example with supplemental figure EV8.

- Fig 6G: What is the evidence that the different regions belong to anti-SARS-CoV-2 specific antibodies? Can this be stated as is even without performing pull-down experiments?

Indeed, it is not possible to state that the antibodies are specific to SARS-CoV-2. With MS-based proteomics we detect the antibody regions available in public databases. However, we cannot find the paratopes themselves. We clarified this now in the text:

“Note that our MS-based proteomics workflow identifies several peptides per immunoglobulin, sufficient to assign them to immunoglobulin regions while not revealing their complete sequence, hence the antigen-binding sites are not covered by this analysis.”

Referee #1 (Remarks for Author):

This is a very comprehensive study of a COVID19 related cohort and I am very positive to seeing this published in EMBO Molecular Medicine after performing additional data analyses that will lift this work beyond the currently existing "reports". There is an opportunity to address heterogeneity in disease course and to include variability as a give factor rather making this an observation that will be seen as a limitation of the outcome.

Referee #2 (Remarks for Author):

In this study Geyer and colleagues perform an extensive proteomic characterisation of the changes in ~300 serum proteins levels for 31 COVID-19 patients over ~30 days which could be compared with serum protein levels of 262 PCR-negative controls. This allowed the authors to study the changes in protein levels between patients and controls as well as the changes in protein levels along the course of the disease. Notable changes include serum proteins linked with innate immunity, regulation of coagulation and lipid homeostasis. By analysing a large number of samples the authors could show that there are clear patterns of co-regulation of the serum proteins across samples, which is expected given that there are multiple groups of functionally linked proteins profiled. Finally they could show that the proteomics results are also capable of reporting on seroconversion with several immunoglobulins and other proteins showing strong correlations with antibody test results and a large number of immunoglobulin regions showing varied patterns of changes in expression.

The study of the immune response and the identification of biomarkers of COVID-19 response continue to be of critical importance. This study is most related with the work by Demichev and colleagues, cited in this manuscript, that has performed a similar analysis of 139 patients, measuring ~300 serum proteins also along the time course of infection. The two studies reach similar conclusions in regards to several of the serum proteins changing upon infection. However, the Demichev study goes further in making use of the data to define biomarkers of response severity. Overall, I find the current study presented here to be very descriptive and not making the best possible use of the data generated (see below). While a prior similar study could sometimes be grounds for suggesting a manuscript for rejection, in this case the work by Demichev et al is itself still under review and more importantly than that, the current study presents an independent cohort that is profiled at very high standards. With some improvements on the analysis I think this work can be of very high value for the study of COVID-19.

Major concerns:

1 - The major concern I have with this study is the very descriptive nature. It would be very important to show that the proteomic data can be used as a predictor of infection outcome. This could include but not be limited to: predicting that a patient is indeed infected (with COVID-19); predicting the severity of the outcome; predicting the time for recovery, etc. For this the authors should make use of the data to build predictors, leaving out some of the their patient data for testing (i.e. machine learning with cross-validation, training/testing). It is important to take into account the characteristics of the patients such as age and other factors that may contribute independently for disease progression.

We thank the Reviewer for the constructive comments to further improve the manuscript. We note that predicting disease severity and outcome was not the primary aim of this study, in particular, because the numbers of patients for such predictions make them only poorly powered. Our main aim was rather to improve our understanding of the pathophysiology of COVID-19 by determining the relevant protein perturbations and find underlying reasons for these alterations.

Nevertheless, we are grateful to the reviewer to push us in this direction. It turns out that the study was indeed adequately powered for a prediction of whether a patient has COVID-19 or not based on the plasma proteome. To this end, we have performed extensive additional analyses and added a new main panel Fig 2D and the supplemental figure Fig EV4A-C. Please see our comments to Reviewer #1 for details. As suggested, we also included the patient characteristics age and sex for the predictions.

Unfortunately, the numbers for disease outcome were too small to draw firm conclusions from machine learning metrics. To illustrate this, when taking the 31 COVID-positive patients comprising of 25 survivors and 6 fatalities when applying a 0.8 – 0.2 train/test split, one would test on approx. 6 samples, with on average one fatality. Only one misclassification change the accuracy by 1/3. Hence, we would prefer to not supply such an analysis here.

2 - The patient cohort is not well describe in terms of age, gender, disease progression and clinical markers. As described in point 1, some of this information needs to be considered when building the predictors as these may confound some of the associations found. That is, there may be serum protein markers that relate to age and not necessarily with disease progression. Such information should be also provided in supplementary materials or if there any privacy concerns through some data sharing mechanism that protects the patient information.

We thank the Reviewer for this suggestion. We have now added patients' characteristics as part of the new supplemental Table EV1. The two groups of COVID-19 patients and PCR-negative controls are very similar with respect to the age distribution with 70.0 ± 14.2 years and 69.5 ± 18.3 years. In the revised version of the manuscript we report in Tables EV2-4 all proteins that are affected by weight loss according to a previous weight loss study (for details see comments to Reviewer #1) and proteins affected by age or gender according to the data in this study.

3 - In my view, the most important benefit of this dataset and analysis will be as independent analysis/cohort from that described in Demichev et al. In an ideal world the predictive models and biomarkers developed in one cohort would be then tested with the independent measurements from the other cohort. This would be fantastic also as it would allow the authors to see the impact of measurements done in different labs etc. Unfortunately, the authors of Demichev et al. did not make the data available in the preprint. In an ideal scenario it would a fantastic service for the community if the authors of this manuscript could reach out to the authors of Demichev et al. to obtain the data for this cohort in attempt to perform a comparative analysis. In the absence of this, it would at least be important to improve the comparison between this manuscript and the one by Demichev et al.

Along with the Reviewer's suggestion, we considered a machine learning prediction for disease outcome. Unfortunately, this turned out to be underpowered due to the small group size of 25 survivors vs. 6 non-survivors. Therefore, we only included a t-test analysis which allowed us to find one strong significant outlier, namely the protein ITIH4. Reassuringly, Völlmy et al. just published a longitudinal study with less time points, but with a similar number of patients. Herein, they were able to identify the same protein, which was differently abundant between survivors and non-survivors. As this study is more similar to our analysis, we used this comparison and included now a statement in our manuscript. **“Reassuringly,**

ITIH4 has been identified as a potential predictor for COVID-19 mortality in an independent study (Völlmy *et al.*, 2021).”

As pointed out before, our main aim was to improve our understanding of COVID-19 pathophysiology and we have not primarily aimed for disease prediction. Nevertheless, increasing the comparability between studies is highly valuable in our point of view. Therefore, we extracted proteins that were longitudinally altered from a supplementary figure of Demichev *et al.* In total 59 out of 89 proteins longitudinally altered in Demichev *et al.*, also changed in our study. We now highlight the proteins in the Tables EV4-6. Even though the majority of proteins overlapped, we want to highlight that different statistical analysis were used.

Minor comments

1 - There is not information in supplementary tables regarding the proteomics dataset itself. This needs to include at least the protein intensities collected for each individual.

We provide this information only upon request due to ethical data protection considerations. For this purpose we uploaded the data on PRIDE.

The MaxQuant output files of the searches have been deposited at the ProteomeXchange Consortium via the PRIDE partner repository and are available via ProteomeXchange with identifier PXD024137.

Username: reviewer_pxd024137@ebi.ac.uk

Password: RlxWqEuA

Due to ethical considerations, we would like to make the MS raw data and MaxQuant output files available for researchers on request and do not permanently deposit them on PRIDE. We added now an according statement in the revised manuscript.

“The MS raw data and MaxQuant output files of the searches generated during and/or analyzed during the current study are available from the corresponding authors on reasonable request.”

2 - At many points the article is very descriptive but does not provide much context on why some information is reported. It would be useful to better connect the observed findings with underlying biology context.

Thank you for this suggestion. We tried to do this now throughout the manuscript at several occasions.

Referee #3 (Remarks for Author):

The manuscript by Geyer et al. describes a large scale MS based biomarker discovery study of COVID-19 patient serum samples. The study made use of symptomatic controls as well as longitudinal samples from 31 COVID-19 patients over an average of 31 days, and by comparing the proteomes of the two cohorts was able to demonstrate specific alterations to the proteome of COVID-19 patient serum samples. The authors also compared the COVID-19 patient samples longitudinally, revealing three major clusters of co-regulated proteins indicating that proteome undergoes complex reorganization over the course of patient hospitalization. In addition to this, the authors demonstrated that a cluster of proteins mostly composed of different immunoglobulin regions correlated with seroconversion as measured by clinical chemistry assays for 5 anti-SARS-COV-2 antibodies. The longitudinal expression of this cluster appeared to be patient specific, and some of the proteomic changes associated with seroconversion were apparent in patients with otherwise negative clinical chemistry results for SARS-COV-2 antibodies suggesting MS based approaches might be more sensitive. From a technical perspective the study was well conducted utilizing a relatively quick (21 minute method) LC-MS method to analyze 60 samples a day, with an average of approximately 310 proteins quantified by LFQ per sample (502 total), demonstrating the high degree of efficiency and sensitivity afforded by the latest LC-MS instrumentation (Evosep One and Bruker timsToF Pro operated in DDA PASEF mode). The paper was generally well written, although some sections could use revision to improve accessibility.

- The work is impressive in its scope and is interesting from a mechanistic perspective, however the relevance of the study to the broader medical community is unclear since few associations were made with disease severity, hospital length of stay, outcome etc. With the exception of the brief discussion of ITIH4, no association was made between the proteomics data and outcome in a predictive sense. Additionally there is no association of the proteomic data with any information concerning disease management, or other patient treatment which might affect the interpretation of the results. For example in figure 6H, it would be interesting to know if medical history or treatment could explain the immunoglobulin profile for patient 22.

We thank Reviewer #3 for the in-depth evaluation of our manuscript and the overall positive assessment. Please see above in the answers to Reviewers 1 and 2 that we included now a machine learning prediction of the disease state (COVID-19 vs. no-COVID-19). In accordance with the Reviewers' suggestions, we also supply additional patient characteristics. This also includes data on immunosuppressive treatment, which was the case for 13 of the COVID-19 patients. However, patient 22 was the only patient showing such an immunoglobulin profile and therefore the treatment might not explain the observation.

- Similarly, a large part of manuscript details the complex longitudinal remodeling of the proteome during hospitalization. It is suggested that these alterations correlate with disease progression, but it would also be interesting to note if any proteins or protein clusters correlate with disease management.

Please see the answer to Reviewer #1. We would like to focus on disease pathophysiology in this manuscript to understand longitudinal disease trajectories. Unfortunately, data on disease management were not available.

- The qualifications for the statistical significance of a putative biomarker are not clearly defined.
- Additionally there is no attempt at validation of any putative marker using complementary methods (MS or otherwise). In some cases clinical chemistry assays correlated (for example CRP), but it would be interesting to validate some of the interesting putative biomarkers (ITIH4) using a more quantitative approach (for example MRM or PRM with stable isotope labeled internal standard peptides).

A true biomarker will have to be confirmed by additional cohorts. As such our aim was to identify biological insights and deliver biomarker candidates that can be followed up as diagnostic markers or open even the possibility for serving as therapeutic targets. This has now been clarified in the manuscript by naming coagulation proteins as potential therapeutic targets as coagulopathies are one of the most frequent complications in COVID-19:

“The description of the detailed regulation of various proteins involved in the coagulation system might even open up the possibility for the development of potential therapeutic avenues.”

We also see high value in validating findings in our manuscript such as ITIH4 or other regulated proteins. We hope that the Reviewer agrees that investigating an additional cohort is out of scope for the current manuscript. Nevertheless, confirming our results is definitely feasible. Please see above the answer to the comment 3 of Reviewer #2 stating that we compared the longitudinal altered proteins with the study of Demichev et al. and that we reference all proteins in supplemental Tables EV4-6 accordingly. Gratifyingly, in the case of ITIH4, in another very recently published manuscript on medRxiv, Völlmy et al. confirmed ITIH4 in a similar study design as a potential marker for morbidity in COVID-19, which we also included with a statement in the revised manuscript.

Specific points for the author's are detailed below:

- The graphical abstract could be simplified. Why are protein trajectories differentiated from proteome alterations? There is no extensive discussion of patient resolution provided in the manuscript.

We supply a simplified version now and are open to choose either of the two version upon editorial decision.

- The abstract states that biomarkers are needed for COVID-19, but this should be clarified. What types of biomarkers for example, and which - if any putative markers were found in this study.

In the revised manuscript, we rephrased the first sentence of the abstract:

“A deeper understanding of COVID-19 on human molecular pathophysiology is urgently needed as a foundation for the discovery of new biomarker and therapeutic targets.”

- In figure 1C, it looks like most proteins detected are clinically utilized biomarkers based on the color coding, but this is not the case.

In total, 71 of the 502 quantified proteins are biomarkers. To avoid misunderstandings, we note the numbers of the biomarkers and the numbers of quantified proteins in the figure and used the same color for clinical applied biomarkers and non-biomarker proteins.

- On page 6, for consistency the expression of CRISP3 should be written as COVID-19+ relative to the control group (ie most down-regulated in COVID-19 samples).

This is now clarified in the text on page 6.

- What (if any) fold change cut-off was used for statistical significance, and how was this determined. Additionally are p-values in the volcano plots (for example figure 2) adjusted for FDR?

We did not use any fold-change cut-off as even very small fold-changes can be interesting if they are highly significant. However, we supply all fold-changes for all statistical tests in the supplemental tables. The p-values of the t-tests are displayed in the figure and the highlighted proteins are significant after adjustment for multi-hypothesis testing with an FDR.

- For the global correlation map (Fig 5) it is not clear from Table EV10 which are the 19 clinical chemistry parameters, it would be good to have this as a separate list. Reference to figure 5C should also include reference to figure EV3? How was the number of correlation coefficients calculated on page 12 (135,460)?

We now highlighted the clinical parameters in an extra list within Table EV10. The correlation coefficients are Pearson correlation coefficients. We also wrote this now in the figure legend and not only in the figure itself.

- In figure 6F, is the x-axis showing the number of patients, or the patient number? If the later could you comment on the apparent trend (decreasing IGs from patient 1 -> 31)

The x-axis in Fig 6F shows the patient number. The trend is depicted by the red bars. We tried to clarify both by new labels for the x-axis and the figure legend.

- Regarding the sample preparation on page 18, samples are referred to as plasma, but they were actually serum in this case.

Thank you very much for finding this typo. We corrected it in the revised version.

- Regarding the conditions for LC-MS/MS, could you comment on the relatively large precursor isolation widths used? What were the accepted mass error tolerances (precursor and product) for peptide assignment and was there a minimum number of unique peptides needed for protein quantitation?

The mass isolation windows used are based on the standard settings suggested by the vendor of the instrument. These may be different from the standard values used with other instrumentation. We added the following statement in the Materials and Methods section:

“The mass tolerance used for the main search of each precursor was set to 20 ppm and the minimum number of peptides needed for a quantification of a protein was set to 1.”

- Regarding the Data Analysis section on page 18, why are the fixed/variable modifications listed specifically for the contaminant database? Should this be for both the reference and contaminant databases?

We thank the reviewer for identifying this issue. Indeed, it should be for both the reference and the contaminant database. We made this more clear now in the Material and Method section.

References

- Bedock, D. *et al.* (2020) 'Prevalence and severity of malnutrition in hospitalized COVID-19 patients', *Clinical Nutrition Espen*, 40, pp. 214–219. doi: 10.1016/j.clnesp.2020.09.018.
- Chen, T. and Guestrin, C. (2016) 'XGBoost: A Scalable Tree Boosting System', in *Proceedings of the 22nd ACM SIGKDD International Conference on Knowledge Discovery and Data Mining. KDD '16: The 22nd ACM SIGKDD International Conference on Knowledge Discovery and Data Mining*, San Francisco California USA: ACM, pp. 785–794. doi: 10.1145/2939672.2939785.
- Di Filippo, L. *et al.* (2020) 'COVID-19 is associated with clinically significant weight loss and risk of malnutrition, independent of hospitalisation: A post-hoc analysis of a prospective cohort study', *Clinical Nutrition*. doi: 10.1016/j.clnu.2020.10.043.
- Dodig-Crnković, T. *et al.* (2020) *Facets of individual-specific health signatures determined from longitudinal plasma proteome profiling*. preprint. *Biochemistry*. doi: 10.1101/2020.03.13.988683.
- Geyer, Philipp E. *et al.* (2016) 'Plasma Proteome Profiling to Assess Human Health and Disease', *Cell Systems*, 2(3), pp. 185–195. doi: 10.1016/j.cels.2016.02.015.
- Geyer, Philipp E *et al.* (2016) 'Proteomics reveals the effects of sustained weight loss on the human plasma proteome', *Molecular Systems Biology*, 12(12), p. 901. doi: 10.15252/msb.20167357.
- Torun, F. M. *et al.* (2021) 'Transparent exploration of machine learning for biomarker discovery from proteomics and omics data', *bioRxiv*, p. 2021.03.05.434053. doi: 10.1101/2021.03.05.434053.
- Völlmy, F. *et al.* (2021) 'Is there a serum proteome signature to predict mortality in severe COVID-19 patients?', *medRxiv*, p. 2021.03.13.21253510. doi: 10.1101/2021.03.13.21253510.

Thank you for the submission of your revised manuscript to EMBO Molecular Medicine. We have now received the enclosed report from the three referees who were asked to re-assess it. As you will see the referees are now supportive and I am pleased to inform you that we will be able to accept your manuscript pending the following amendments:

1. In the main manuscript file, please do the following.

***** Reviewer's comments *****

Referee #1 (Comments on Novelty/Model System for Author):

The authors have addressed most of my previous concerns in a sufficient manner and provided a revised version that was certainly improved. The changes made could have been more extensive and the section on the serology comparison could have been shortened to keep a focus on the proteome trajectories.

The work elegantly illustrated that proteome changes in response to SARS-CoV-2 infections occur first for inflammatory proteins, then those related to coagulation and metabolism. Liver proteins such as ITHs and APOs appear as important indicators of disease or severity thus offer a diagnostic utility, thus liver injury could serve as a clinical trait to monitor progression and recovery.

Referee #1 (Remarks for Author):

Remarks for Author: The authors have addressed most of my comments in a sufficient manner. More biomedical insights could have been lifted but the overall focus on proteomics data is still providing a valuable contribution to our growing knowledge about the pathophysiology of COVID19.

Referee #2 (Remarks for Author):

The authors have addressed most of the concerns raised previously.

In regards to data sharing it is quite unfortunate that the authors believe that this data should not be made available as this diminishes the value of this work. I will let such matters up to the editorial guidelines and practices of the journal but I would encourage the authors to find a mechanism for data sharing that does not depend on a request to the authors.

Referee #3 (Remarks for Author):

I have reviewed the revised manuscript and I am satisfied with the changes that the authors have made and would now recommend it for publication.

The authors have made all requested editorial changes.

We are pleased to inform you that your manuscript is accepted for publication and is now being sent to our publisher to be included in the next available issue of EMBO Molecular Medicine.

Corresponding Author Name: Philipp E Geyer, Daniel Teupser

Manuscript Number: EMM-2021-14167